# Coordinating Distributed Example Orders for Provably Accelerated Training

**A. Feder Cooper**\*  **Wentao Guo**\*  **Khiem Pham**\*

**Tiancheng Yuan**  **Charlie F. Ruan**  **Yucheng Lu**  **Christopher De Sa**

Cornell University
{afc78, wg247, dkp45, ty373, cfr54, yl2967, cmd353}@cornell.edu

## Abstract

Recent research on online Gradient Balancing (GraB) has revealed that there exist permutation-based example orderings for SGD that are guaranteed to outperform random reshuffling (RR). Whereas RR arbitrarily permutes training examples, GraB leverages stale gradients from prior epochs to order examples — achieving a provably faster convergence rate than RR. However, GraB is limited by design: while it demonstrates an impressive ability to scale-up training on *centralized* data, it does not naturally extend to modern *distributed* ML workloads. We therefore propose *Coordinated Distributed GraB* (CD-GraB), which uses insights from prior work on kernel thinning to translate the benefits of provably faster permutation-based example ordering to distributed settings. With negligible overhead, CD-GraB exhibits a linear speedup in convergence rate over centralized GraB and outperforms distributed RR on a variety of benchmark tasks.

## 1 Introduction

Random reshuffling, which samples training-data examples without replacement, has become the *de facto* example-ordering method in modern deep-learning libraries [34], given that it tends to accelerate optimizer convergence in practice. However, some recent theoretical work has identified cases in which random reshuffling can lead to data orderings that have a poor effect on convergence [8, 36, 49]. This has encouraged a line of research to investigate if there exist provably better permutation-based orderings that afford greater scalability in training [22, 24, 31]. Notably, Lu et al. [24] connects permuted-order SGD to the *herding problem* [16], and proposes the herding-based online Gradient Balancing algorithm (GraB), which converges provably faster than random reshuffling, and does so with little memory or computational overhead. In fact, in follow-on work, Cha et al. [6] proves that GraB is optimal: in theory, GraB is the fastest possible permutation-based example ordering algorithm.

These results are very exciting, suggesting that GraB should unseat random reshuffling as the example ordering method-of-choice for SGD; however, they only hold with respect to a *single* machine. GraB is optimal in settings with *centralized* data, but does not naturally translate to problems of modern-ML scale, which demand that training workloads be distributed across *multiple parallel* workers that each only have access to a subset of the training data. This drawback raises an important question:

> *Can we simultaneously achieve the scalability benefits of distributed training and*
> *provably faster permutation-based example ordering for SGD — both in theory and in practice?*

In this work, we show that it is indeed possible to attain these twin objectives. To do so, we suggest the online **C**oordinated **D**istributed **Gra**diant **B**alance algorithm (CD-GraB), which leverages insights

37th Conference on Neural Information Processing Systems (NeurIPS 2023).

---

\*Equal contribution

from kernel thinning to elevate the herding framework of centralized GraB (GraB) to the parallel setting. Felicitously, as a side effect, this choice of formulation brings about positive practical performance benefits (that can also improve the empirical behavior of centralized GraB). Using the exact same assumptions as the original GraB paper, **we show analytically that coordinating example orders across parallel workers leads a linear speedup in convergence rate**. For $T$ epochs and $m$ parallel workers, each with access to $n$ examples, CD-GraB's convergence rate is $\tilde{O}((mnT)^{-2/3})$ on smooth, non-convex objectives and $\tilde{O}((mnT)^{-2})$ under the Polyak-Łojasiewicz (P.L.) condition.[2]

We run a series of experiments to verify these improvements in practice, implementing CD-GraB on a single node that distributes computation across multiple GPUs. We also run an ablation study in order to disentangle the benefits of parallelism from the positive side effects of using kernel thinning to formulate the CD-GraB algorithm. Similar to how centralized GraB demonstrates improved generalization over centralized random reshuffling (RR), we observe that CD-GraB exhibits improved generalization over distributed random reshuffling (D-RR). Altogether, the success of our work suggests a new distributed training paradigm to explore in future work, which we call the *Order Server* (Section 6). In summary, we:

- Propose the online **C**oordinated **D**istributed **Gra**dient **B**alancing (CD-GraB) algorithm, which enables provably accelerated training using SGD in the parallel setting (Section 3);

- Prove that the convergence rate for CD-GraB exhibits a linear speedup over GraB, using the exact same assumptions as the original GraB paper (Section 4);

- Produce extensive empirical validation of CD-GraB's improved scalability on a variety of tasks in deep learning and on large-scale logistic regression (Section 5).

## 2 Preliminaries and Related Work

In this section, we discuss the preliminaries and prior scholarship on permutation-based example ordering, with particular attention paid to the centralized online Gradient Balancing Algorithm (GraB) [24]. This lays the groundwork for how our coordinated, distributed GraB algorithm (Section 3) imparts the efficiency guarantees of GraB to the parallelized regime (Section 4).

**Ordering data examples during training.** Training a model can be formulated as minimizing a differentiable loss function $f : \mathbb{R}^d \to \mathbb{R}$ over $N$ data examples. The goal of this minimization is to obtain the target model weights $\boldsymbol{w}^* = \arg\min_{\boldsymbol{w}} f(\boldsymbol{w})$, where $f(\boldsymbol{w}) = \frac{1}{N} \sum_{j=1}^{N} f(\boldsymbol{w}; j)$, for which $f(\boldsymbol{w}; j)$ denotes the loss incurred on the $j$-th example. A typical training process iteratively updates the model parameters $\boldsymbol{w}$ by scanning over the $N$ data examples repeatedly, with $t$-th scan (or epoch) following

$$\boldsymbol{w}_t^{j+1} = \boldsymbol{w}_t^j - \alpha \nabla f(\boldsymbol{w}_t^j; \pi_t(j)), \ \ \forall j \in [N], \tag{1}$$

where $\alpha$ denotes the learning rate, and $\pi_t : [N] \to [N]$ denotes a permutation ordering[3] adopted in the $t$-th epoch from which the examples are chosen to compute gradients, $\boldsymbol{w}_t^1$ denotes the initial model weights for the $t$-th epoch, and $\boldsymbol{w}_t^j$ denotes the model weights after $j-1$ gradient updates in the $t$-th epoch.[4]

The choice of ordering $\pi$ can have a significant effect on optimizer performance. Two popular methods, which can demonstrate convergence speedups in practice, are 1) random reshuffling (RR) [46], for which the permutations are random and differ over epochs, and 2) Shuffle Once (SO) [4, 13], for which a random permutation is computed once and remains fixed for all epochs. Recht and Ré [37] conducted the first theoretical investigation of RR, while subsequent works like Yun et al. [49] and De Sa [8] have given counterexamples in which RR leads to orderings that have a poor effect on convergence. Altogether, many studies indicate that RR and SO only provide efficiency benefits under certain conditions [14, 15, 30].

These limitations of RR and SO have motivated research to identify permutations that outperform random ones. Rajput et al. [36] introduces an RR variant that achieves improved convergence for quadratics by reversing the ordering every other epoch. Other non-RR-based methods pick efficient orderings based on correlations between adjacently selected examples. In a recent line of work, Lu et al.

---

[2]In this paper, we use $\tilde{O}$ by convention to hide logarithmic factors in the problem parameters.

[3]While without-replacement orderings are most common in large-scale learning [5], ordering strategies need not be permutations, e.g., with-replacement sampling [23, 32, 39] or curriculum learning [12, 27, 41].

[4]Note that we write (1) in terms of per-example-$j$ gradients.

[22] proves that faster convergence is possible for SGD when the averages of consecutive stochastic gradients converge faster to the full gradient. Based on this result, in follow-on work Lu et al. [24] proposes the centralized online Gradient Balancing algorithm (GraB), which outperforms RR, and upon which we base this work.

## 2.1 GraB: Optimal, online, permutation-based example ordering for centralized ML

GraB is a permutation-based example-ordering algorithm that identifies provably better-than-random orderings *in centralized, single-node settings* for SGD. GraB finds such orderings by leveraging information in stale stochastic gradients from previous epochs to guide ordering in the next epoch. More formally, for smooth, non-convex objectives, Lu et al. [24] proves that any permutation $\pi^*$ that guarantees

$$\max_{k \in [N]} \left\| \sum_{j=1}^k \nabla f(\boldsymbol{w}; \pi^*(j)) - \nabla f(\boldsymbol{w}) \right\|_\infty = \tilde{O}(1) \quad (\nabla f(\boldsymbol{w}) \text{ is the average gradient}), \quad (2)$$

will yield a convergence rate of $\tilde{O}((NT)^{-2/3})$ (for epochs $T$) for SGD, which is superior to the $O(N^{-1/3}T^{-2/3})$ convergence rate of random reshuffling [30].

**GraB's connection to herding and balancing.** To find such a permutation $\pi^*$, Lu et al. [24] connect (2) to the *herding problem* and vector *balancing* [16, 45]. Understanding why GraB does not naturally extend to the distributed setting — and our main contributions (Sections 3 and 4) — requires some additional details on the fundamentals of herding:

Given $N$ vectors[5] $\{\boldsymbol{z}_j\}_{j=1}^N$ ($\boldsymbol{z}_j \in \mathbb{R}^d$), $\|\boldsymbol{z}_j\|_2 \le 1$ ($\forall j$), herding identifies a permutation $\pi^*$ such that

$$\max_{k \in [N]} \left\| \sum_{j=1}^k \left( \boldsymbol{z}_{\pi^*(j)} - \bar{\boldsymbol{z}} \right) \right\|_\infty = \tilde{O}(1), \quad \text{where } \bar{\boldsymbol{z}} = \frac{1}{N} \sum_{j=1}^N \boldsymbol{z}_j. \quad (3)$$

It is clear that (3) generalizes (2), which is a specific case of herding in an optimization setting.

Harvey and Samadi solve (3) with a method called *balancing* [16]. Balancing uses a *signed* version of the herding problem to optimize any given permutation $\pi$ to reduce the bound in (3). That is, balancing formulates the signed herding problem

$$\max_{k \in [N]} \left\| \sum_{j=1}^k s_{\pi(j)} \left( \boldsymbol{z}_{\pi(j)} - \bar{\boldsymbol{z}} \right) \right\|_\infty, \quad \text{where } \{s_j\}_{j=1}^N \in \{+1, -1\}. \quad (4)$$

Given a group of such signs $\{s_j\}_{j=1}^N$ and an arbitrary permutation $\pi$, Harvey and Samadi prove that Algorithm 1 produces a new permutation $\pi'$ such that

$$\max_{k \in [N]} \left\| \sum_{j=1}^k \left( \boldsymbol{z}_{\pi'(j)} - \bar{\boldsymbol{z}} \right) \right\|_\infty \le \frac{1}{2} \max_{k \in [N]} \left\| \sum_{j=1}^k s_{\pi(j)} \left( \boldsymbol{z}_{\pi(j)} - \bar{\boldsymbol{z}} \right) \right\|_\infty + \frac{1}{2} \max_{k \in [N]} \left\| \sum_{j=1}^k \left( \boldsymbol{z}_{\pi(j)} - \bar{\boldsymbol{z}} \right) \right\|_\infty.$$

This says that, with new permutation $\pi'$, the objective of (3) now approaches the bound of (4). Importantly, recent advances show that it is quite cheap to find a group of signs, such that (4) is on the order of $\tilde{O}(1)$ (e.g., Alweiss et al. [2], in Algorithm 2). We are therefore able to call Algorithm 1 repeatedly, which will eventually obtain the $\pi^*$ that solves the $\tilde{O}(1)$ herding objective in (3).

---

**Algorithm 1** Reordering Vectors based on Balanced Signs [Harvey and Samadi [16]]

---

**input:** a group of signs $\{s_j\}_{j=1}^N$, initial order $\pi$
**initialize:** two order-sensitive lists $L_{\text{pos}} \leftarrow []$, $L_{\text{neg}} \leftarrow []$.
**for** $j = 1...N$ **do**
  $L_{\text{pos}}$.append($\pi(j)$) **if** $s_j$ is $+1$ **else** $L_{\text{neg}}$.append($\pi(j)$).
**end for**
**return:** new order $\pi' := \text{concat}(L_{\text{pos}}, \text{reverse}(L_{\text{neg}}))$.

---

**GraB's application of herding to gradient balancing.** Lu et al. [24] applies this framework of herding and balancing to develop GraB, i.e., to minimize (2). The main challenge for the success of this approach is to find the right gradients $\boldsymbol{z}_j$ in the optimization context of (2). Notably, the herding and balancing framework requires the vector mean $\bar{\boldsymbol{z}}$ in advance. To satisfy this requirement, GraB "centers" the gradient vectors using a *stale mean*. That is, GraB runs the herding algorithm on vectors that are defined as

$$\boldsymbol{z}_j = \nabla f(\boldsymbol{w}_t^j; \pi_t(j)) - \frac{1}{N} \sum_{p=1}^N \nabla f(\boldsymbol{w}_{t-1}^p; \pi_{t-1}(p)), \quad (5)$$

where $\boldsymbol{w}_t^p$ denotes the model weights after $p - 1$ updates in the $t$-th epoch, and $\pi_t$ denotes the permutation adopted in the $t$-th epoch. Lu et al. [24] proves that this definition of $\boldsymbol{z}_j$ preserves the

---
[5]Herding does not have an optimization context. Here, $N$ does *not* refer to the number of data examples used in training [1]; rather, $N \in \mathbb{Z}^+$ describes the size of a set of arbitrary vectors. We slightly abuse notation because we execute the herding subroutine on exactly $N$ gradients (Section 3), which happen to equal the number of $N$ examples.

benefits of balancing with negligible noise or overhead. The only overhead comes from storing the running average of the gradients in epoch $t-1$ to "center" the gradients in the subsequent epoch $t$.

With this approach, Lu et al. [24] proves that GraB demonstrates more efficient convergence than RR for SGD. Better still, Cha et al. [6] demonstrates that GraB is in fact the *optimal* permutation-based ordering method for SGD: it is not possible to produce a permutation-based ordering in the centralized setting that achieves a faster convergence rate for SGD.

Despite GraB's clear benefits over RR, it assumes local access to all examples. This assumption does not hold for popular, modern, parallel settings (e.g., parameter server [20]), in which workers only have access to subsets of examples. No present work has attempted to investigate GraB's applicability to this setting. While some work has studied distributed RR (D-RR) [18, 26, 38, 48], it remains an open question if GraB's efficiency benefits for SGD can be conferred to the modern-scale, distributed-ML setup.

## 3 CD-GraB: A Provably Efficient Ordering Algorithm for Distributed Training

Our main contribution is to elevate GraB to the parallel regime, so that distributed training can enjoy the efficiency benefits of provably better example ordering. Based on the preliminaries, we can now explain why this is not a straightforward task: **While GraB achieves the optimal convergence rate for SGD on centralized data, it does not naturally translate to a distributed setting** (Section 3.1). Our key insights for resolving these problems are to reformulate the herding framework in Lu et al. [24] to work in parallel, and to leverage insights from kernel thinning [3, 10, 11] to derive the *online* PairBalance algorithm, which solves this parallelized herding objective (Section 3.2). Lastly, we present the full-stack CD-GraB algorithm that makes our solution work in practice (Section 3.3). The server implements online PairBalance, which coordinates gradient information from the distributed workers in training epoch $t$ in order to determine a provably efficient example order for the next epoch $t+1$ (Section 4).

### 3.1 Issues with GraB in the distributed setting

To clarify the issues with distributing GraB, we first need to define the distributed training setup more precisely. We consider the standard data-parallel regime with $m$ parallel workers, where each worker keeps a copy of the model weights $\boldsymbol{w} \in \mathbb{R}^d$ and maintains $n = N/m$ local examples.[6] As in many data-parallel training applications,[7] such as geo-distributed model training [47], we assume *the data examples cannot be shared or moved across workers*. More formally, this setup can be expressed as

$$\min_{\boldsymbol{w} \in \mathbb{R}^d} \left[ f(\boldsymbol{w}) = \frac{1}{m} \sum_{i=1}^{m} f^i(\boldsymbol{w}) \right] \quad \text{with} \quad f^i(\boldsymbol{w}) = \frac{1}{n} \sum_{j=1}^{n} f^i(\boldsymbol{w}; j), \tag{6}$$

where $f^i(\boldsymbol{w}; j) : \mathbb{R}^d \to \mathbb{R}$, $j \in [n]$, denotes the loss incurred on the $j$-th example on the $i$-th worker for model weights $\boldsymbol{w}$. We can now consider running (1) using this setup, for which each worker scans over their $n$ local-data examples using (potentially) different permutations. We denote $\pi_{t,i} : [n] \to [n]$ as the permutation-based ordering adopted on the $i$-th worker in the $t$-th training epoch. Adjusting (1) to accommodate the setup in (6), the update to the model can be summarized as

$$\boldsymbol{w}_t^{j+1} = \boldsymbol{w}_t^j - \frac{\alpha}{m} \sum_{i=1}^{m} \nabla f^i(\boldsymbol{w}_t^j; \pi_{t,i}(j)), \ \forall j \in [n]. \tag{7}$$

That is, in epoch $t$, each worker $i$ selects their respective, local $j$-th example according to $\{\pi_{t,i}\}_{i=1}^n$ in order to compute stochastic gradients (Appendix).

**Following this setup, Algorithm 1 no longer guarantees the $\tilde{O}(1)$ bound to the herding problem (3)**, a bound that is valid only when *all* data examples can be permuted *freely* [16]. This constraint is fine for centralized GraB, but, in distributed training, workers only have access to a *subset* of examples. Distributed training requires that *worker-specific permutations only involve the examples in their respective local subsets*. Further, recall that GraB uses stale means to center gradients (5) in order to solve the herding objective. This, too, causes problems in distributed training. In practice, it is typical to employ larger learning rates $\alpha$ for greater scalability [40]; larger $\alpha$ increases the discrepancy between averaged gradients in adjacent epochs, which, in turn, would make GraB's use of stale means unreliable.

---

[6]Without loss of generality, we assume the $N$ examples are divided evenly among the $m$ workers and $n$ is even.

[7]One such popular paradigm is federated learning [28, e.g.]. Federated learning typically involves highly imbalanced loads, heterogeneous data, partial user participation, and additional privacy-preserving mechanisms. These characteristics are orthogonal to what we consider here for example order. If we were to allow for such data organization, we would need to assume non-global communication per iteration or additional constraints on how global communication occurs. For CD-GraB, we focus on the regime of using parallelism to accelerate training.

## 3.2 Our efficient solution: parallel herding and pair balancing

To address the limitations presented in the prior section, which preclude the direct application of GraB to distributed training, we will need to **1) reformulate the herding problem to fit the parallel setting, and 2) redesign how to do gradient balancing**, such that it both solves our new herding formulation and allows for reliability with higher learning rates. We now present our solution to both these problems; we introduce the *parallel herding* problem and the online PairBalance subroutine that solves it.

**Parallel Herding.** To extend herding to the parallel setting, consider the following setup: There are $m$ workers, which each have local access to $n$ vectors. Let $z_{i,j} \in \mathbb{R}^d$ denote the vector indexed by $j$ on the $i$-th worker. Assuming $\|z_{i,j}\|_2 \le 1$ $(\forall i \in [m], \forall j \in [n])$, the goal of parallel herding is to find $m$ permutations, $\pi_1, \pi_2, ..., \pi_m$ where $\pi_i : [n] \to [n]$ $(\forall i \in [m])$, so as to minimize:

$$\max_{k \in [n]} \left\| \sum_{j=1}^{k} \sum_{i=1}^{m} \left( z_{i,\pi_i(j)} - \bar{z} \right) \right\|_\infty, \quad \text{with} \quad \bar{z} = \frac{1}{mn} \sum_{i=1}^{m} \sum_{j=1}^{n} z_{i,j}. \tag{8}$$

When directly comparing (8) with (3), it is clear that parallel herding differs in two notable ways from the original herding problem. First, each permutation $\pi_i : [n] \to [n]$ $(\forall i \in [m])$ only decides the ordering of the $n$ vectors that are associated with worker $i$. Second, the prefix sum taken in the objective norm is accumulated over all the workers (the inner sum from $i = 1...m$). This formulation naturally captures the setting in a distributed environment: **workers need to decide permutations collaboratively, and the worker-specific vectors are processed simultaneously rather than sequentially**.

Given that this formulation fits the distributed setting, we next need to show that parallel herding does in fact address the limitations posed by centralized GraB: that it is possible recover the original $\tilde{O}(1)$ herding bound, and that we can solve the issue of unreliable stale gradients (Section 3.1). The solution that we present in the remainder of this section is a new vector balancing subroutine: online PairBalance. To give an intuition, as its name suggests, online PairBalance leverages insights from kernel thinning to *balance* vector differences over vector *pairs*. This also eliminates the need to perform vector centering, and thus solves the stale mean problem.

**Using kernel thinning to solve parallel herding.** We call our solution to the parallel herding objective (8) *pair balancing*, which we derive from key insights in *kernel thinning* [3, 10, 11]. In particular, Dwivedi and Mackey show that it is possible to solve the herding objective in $\tilde{O}(1)$ **by only examining differences on *pairs of examples*** [10]. They derive an algorithm that generalizes Alweiss et al. [2, subroutine in Algorithm 2], which solves herding in $\tilde{O}(1)$ (Section 2), and does so by operating only on vector-pair differences.[8] This comes with a very useful property: eliminating the requirement of knowing the maximum vector norm ahead of time and centering the vectors (i.e., making all the vectors sum to zero) in order to solve the herding problem. This is the key to solving the parallel herding objective (8) in $\tilde{O}(1)$, and elevating the benefits of GraB to a distributed setting.

Following Dwivedi and Mackey [10], we will balance over paired vectors, and will do so in an *online* fashion (Section 3.3). This eliminates GraB's requirement of using a stale mean to center gradient vectors (Section 2.1), but still minimizes the parallel herding objective to $\tilde{O}(1)$. We defer proving this result to Section 4, and first describe our concrete algorithm. Online PairBalance applies Algorithm 1 on the "flattened" and "paired" sequence of all of the workers' paired-difference gradients, i.e.,

$$y_{n(k-1)+i} = z_{i,2k-1} - z_{i,2k}, \quad \forall k \in \left[\frac{n}{2}\right], \quad i = 1...m.$$

That is, we fit these ordered-paired differences $\{y_i\}_{i=1}^{mn/2}$ into the herding and balancing framework (Algorithm 1): if sign $s$ is associated with $y_{n(k-1)+i}$, then $z_{i,2k-1}$ and $z_{i,2k}$ receive $s$ and $-s$, respectively.

## 3.3 The full-stack CD-GraB algorithm

Having solved the parallel herding problem with pair balancing, we now demonstrate how to bring everything together in an optimization context to *coordinate distributed gradient balancing* for distributed training. That is, we can now introduce our full-stack CD-GraB algorithm, which trains models in a distributed setting (Section 3.1) while efficiently ordering the examples by using PairBalance (Section 3.2, Algorithm 2) in an online manner.

---

[8] Dwivedi and Mackey minimize the maximum mean discrepancy (MMD) between a selected coreset and an empirical distribution. They develop a new self-balancing Hilbert walk on differences of *pairs of examples* to select exactly half of the dataset points, and solve coreset selection by iteratively halving the input vector sequence into balanced coresets then selecting and refining a candidate coreset to minimize MMD with the input sequence.

**Algorithm 2** PairBalance

$\triangleright$ The inputs, outputs and subroutine for this algorithm are order-sensitive

**input:** current running sum $r$, paired vectors $z_1, z_2$

**compute:** $s, r \leftarrow \text{RandomizedBalance}(r, z_1 - z_2)$
**return:** $s$ (sign for $z_1$),
$\qquad -s$ (sign for $z_2$),
$\qquad\quad r$ (updated running sum)

$\triangleright$ Adapted from Alweiss et al. [2]
**define subroutine:** RandomizedBalance$(r, c)$
$\quad$**compute:** $p \leftarrow \frac{1 - \langle r, c \rangle}{2}$
$\quad$**compute:** $s \leftarrow +1$ with probability $p$;
$\qquad\qquad\quad s \leftarrow -1$ with probability $1 - p$
$\quad$**update:** $r \leftarrow r + sc$
$\quad$**return:** $s, r$

Server

4. Call Pair Balance: $g_1^1 - g_2^1$, $g_3^1 - g_4^1$, $g_1^2 - g_2^2$, $g_3^2 - g_4^2$
5. Call Algorithm 1
6. New Data Permutations $\pi_{t+1,1}$ $\pi_{t+1,2}$

3. Send Gradients to server

Worker 1 | Worker 2 | Worker 1 | Worker 2 | Worker 1 | Worker 2

$X_1^1$ $X_1^2$ $g_1^1$ $g_1^2$ $X_1^1$ $X_1^2$
$X_2^1$ $X_2^2$ $g_2^1$ $g_2^2$ $X_4^1$ $X_4^2$
$X_3^1$ $X_3^2$ $g_3^1$ $g_3^2$ $X_1^1$ $X_3^2$
$X_4^1$ $X_4^2$ $g_4^1$ $g_4^2$ $X_2^1$ $X_2^2$

1. Examples ordered with $\pi_{t,1}$ $\pi_{t,2}$
2. Compute gradients
7. Reorder examples with $\pi_{t+1,1}$ $\pi_{t+1,2}$

Figure 1: **Left:** The PairBalance algorithm, which the server runs online. **Right:** CD-GraB running on one server (top) and two workers (bottom). The workers do not share data examples.

We describe CD-GraB at two levels of abstraction: a high-level illustration (Figure 1, steps 1–7) and a detailed pair of worker-server algorithm statements (Figure 2). Since the workers only have access to a subset of the training data, in parallel they compute local, per-example stochastic gradients and send them to the server. The server simultaneously calls PairBalance online, which coordinates information from all the workers' gradients (i.e., using adjacent example-specific gradients) to determine the next epoch's worker-specific permutations. In more detail:

In epoch $t$, (Figure 1, step 1) the two workers have permutations $\pi_{t,1}$ and $\pi_{t,2}$, respectively. Each worker computes per-example gradients $g_j^i$ (2; Algorithm 3:4), and sends them to the server (3; Algorithm 3:5). The server we implement functions as a parameter server [20]: It computes the average of the workers' per-example gradients (Algorithm 4:6), and sends it back to all workers (Algorithm 4:7) so that they can update their local models (Algorithm 3:6-7). Simultaneously, as the server receives gradients (Algorithm 4:5), it calls PairBalance (Algorithm 2) on adjacent vectors (4; Algorithm 4:4-13). PairBalance produces signs to supply to the reordering algorithm (Algorithm 1), which, using the current worker permutations $\pi_{t,i}$, produces the new per-worker permutations for the next epoch (5; Algorithm 4:14). In Figure 1, these correspond to $\pi_{t+1,1}$ and $\pi_{t+1,2}$, which the server then sends back to the respective workers (6; Algorithm 4:15). Lastly, before the start of the next epoch, the workers reorder their examples according to the new permutations (7; Algorithm 3:9).

## 4 Convergence Analysis

We next demonstrate formally that our CD-GraB algorithm (Section 3.3) confers the efficiency benefits of centralized GraB (Section 2.1) to the distributed setting. In brief, our main theoretical results show that **CD-GraB enjoys a linear speedup in convergence rate** under two sets of conditions: smoothness (Theorem 2) and the Polyak-Łojasiewicz (P.L.) condition (Theorem 3). **Both results guarantee that CD-GraB is faster than distributed random reshuffling (D-RR)**. Our proofs rely on Corollary 7 from Dwivedi and Mackey [10], which shows that, with high probability, RandomizedBalance (subroutine in Algorithm 2, from Alweiss et al. [2]) guarantees a $\tilde{O}(1)$ bound to the signed herding objective (4).[9]

To begin, we restate this result to cohere with our framework, for which the vectors $z_j$ are gradients in an optimization context:

**Theorem 1** (**Corollary 7, Dwivedi and Mackey [10]**). *Consider any vectors $\{z_j\}_{j=1}^N$ ($z_j \in \mathbb{R}^d$) with $\|z_j\|_2 \le 1$ supplied as input to the* RandomizedBalance *subroutine in Algorithm 2. Then for any $\delta > 0$, with probability at least $1 - \delta$,* RandomizedBalance *outputs a sequence of signs $\{s_j\}_{j=1}^N \in \{-1, 1\}$ that satisfy $\max_{k \in [N]} \left\| \sum_{j=1}^k s_j z_j \right\|_\infty \le \tilde{A}$, where $\tilde{A} = \sqrt{2\log\left(\frac{4d}{\delta}\right)\log\left(\frac{4N}{\delta}\right)} = \tilde{O}(1)$.*

---

[9]Corollary 7 from Dwivedi and Mackey [10] improves the result of Theorem 1.1 from Alweiss et al. [2].

| **Algorithm 3** CD-GraB Workers | **Algorithm 4** CD-GraB Parameter Server |
|---|---|

**Algorithm 3** CD-GraB Workers

**require:** $m$ workers, $n \coloneqq \frac{N}{m}$ ex. per worker
**input:** initial $\boldsymbol{w}_1^1$, epochs $T$, learning rate $\alpha$

1: **receive:** initial permutations $\xleftarrow{\quad \{\pi_{1,i}\}_{i=1}^m \quad}$
2: **for** epoch $t \coloneqq 1...T$ **do**
    ▷ Run in parallel for workers $i = 1...m$
3:     **for** example $j \coloneqq 1...n$ **do**
4:         **compute:** $\boldsymbol{g}_j^i \leftarrow \nabla f^i(\boldsymbol{w}_t^j, \pi_{t,i}(j))$
5:         **send:** $\boldsymbol{g}_j^i \xrightarrow{\quad j\text{-th stochastic grad. } \boldsymbol{g}_j^i \quad}$
6:         **receive:** $\bar{\boldsymbol{g}}_j \xleftarrow{\quad \text{avg. } j\text{-th stochastic grad. } \bar{\boldsymbol{g}}_j \quad}$
7:         **update:** $\boldsymbol{w}_t^{j+1} \leftarrow \boldsymbol{w}_t^j - \alpha \bar{\boldsymbol{g}}_j$
8:     **end for**
9:     **receive:** next permutation $\xleftarrow{\quad \pi_{t+1,i} \quad}$
10:     **update:** $\boldsymbol{w}_{t+1}^1 \coloneqq \boldsymbol{w}_t^{n+1}$
11: **end for**
12: **return:** $\boldsymbol{w}_{T+1} \coloneqq \boldsymbol{w}_{T+1}^1$

**Algorithm 4** CD-GraB Parameter Server

**require:** $m$ workers, $n \coloneqq \frac{N}{m}$ ex. per worker
**input:** epochs $T$

1: **send:** initial permutations $\{\pi_{1,i}\}_{i=1}^m$
2: **for** epoch $t \coloneqq 1...T$ **do**
3:     **initialize:** running sum $\boldsymbol{h} = \boldsymbol{0}$; empty list $\mathcal{S}$
4:     **for** example $j \coloneqq 1...n$ **do**
5:         **receive:** $\{\boldsymbol{g}_j^i\}_{i=1}^m$ from all workers $i$
6:         **compute:** avg. gradient: $\bar{\boldsymbol{g}}_j \leftarrow \frac{1}{m} \sum_{i=1}^m \boldsymbol{g}_j^i$
7:         **send:** $\bar{\boldsymbol{g}}_j$ to all the workers
8:         **for** worker $i \coloneqq 1...m$ **do**
9:             **if** $j \bmod 2 = 0$:
10:             $\boldsymbol{h}, s_{j-1}^i, s_j^i \leftarrow \mathsf{PairBalance}(\boldsymbol{h}, \boldsymbol{g}_{j-1}^i, \boldsymbol{g}_j^i)$
11:             $\mathcal{S}.\mathsf{append}(s_{j-1}^i)$; $\mathcal{S}.\mathsf{append}(s_j^i)$
12:         **end for**
13:     **end for**
    ▷ Call Alg. 1 for $i = 1...m$ on $\pi_{t,i}$ and $\mathcal{S}$
14:     **compute:** next permutations $\{\pi_{t+1,i}\}_{i=1}^m$
15:     **send:** $\{\pi_{t+1,i}\}_{i=1}^m$ to each worker $i$
16: **end for**

Figure 2: CD-GraB worker and server (here, a parameter server [20]) algorithms.

To integrate this result with our parallel setting, we need some additional assumptions that are standard in the literature on distributed optimization — that the variance of the per-example gradients on each worker is uniformly bounded (Assumption 1), and that the variance between worker-specific gradients is similarly bounded (Assumption 2). More precisely, following the distributed setup in (7), we denote the global loss gradient to be $\nabla f(\boldsymbol{w})$, each $i$-th worker's local loss gradient to be $\nabla f^i(\boldsymbol{w})$ ($\forall i \in [m]$), and each $i$-th worker's per-example loss gradients to be $\nabla f^i(\boldsymbol{w};j)$ ($\forall j \in [n]$). We assume:

**Assumption 1** (**Bounded Gradient Variance**). *For all $i \in [m]$ there exists a constant $\sigma > 0$ such that for all $j \in [n]$ and for all $\boldsymbol{w} \in \mathbb{R}^d$, it holds that $\left\| \nabla f^i(\boldsymbol{w};j) - \nabla f^i(\boldsymbol{w}) \right\|_2^2 \leq \sigma^2$.*

**Assumption 2** (**Bounded Data Heterogeneity**). *There exists a constant $\varsigma > 0$ such that $\forall i \in [m]$, $\left\| \nabla f^i(\boldsymbol{w}) - \nabla f(\boldsymbol{w}) \right\|_2^2 \leq \varsigma^2$.*

Lastly, we include one additional assumption from the original GraB paper [24]: we assume a cross norm $L_{2,\infty}$ (which can be easily adapted to $L_2$-smoothness by setting $L_{2,\infty}$ to be $\sqrt{d}L_2$).

**Assumption 3** (**Smoothness**). *There exists constant $L_{2,\infty} > 0$ such that for any $\boldsymbol{w}, \boldsymbol{v} \in \mathbb{R}^d$, any $i \in [m]$, and any $j \in [n]$, it holds that $\left\| \nabla f^i(\boldsymbol{w};j) - \nabla f^i(\boldsymbol{v};j) \right\|_2 \leq L_{2,\infty} \| \boldsymbol{w} - \boldsymbol{v} \|_\infty$.*

Given these assumptions, we can prove a convergence guarantee for CD-GraB:

**Theorem 2.** *Suppose that Assumptions 1, 2 and 3 hold. For any $\delta > 0$, if we set learning rate $\alpha$ to be*

$$\alpha = \min \left\{ \frac{1}{16 L_{2,\infty}(2n + \tilde{A}/m)}, \left( \frac{4 F_1 m^2}{42 L_{2,\infty}^2 (\varsigma + \sigma)^2 \tilde{A}^2 n T + 18 L_{2,\infty}^2 m^2 n^3 \sigma^2} \right)^{1/3} \right\},$$

*where $F_1 = f(\boldsymbol{w}_1) - \inf_{\boldsymbol{w} \in \mathbb{R}^d} f(\boldsymbol{w})$ and $\tilde{A}$ comes from Theorem 1. Then, with probability at least $1 - T\delta$,*

$$\frac{1}{T} \sum_{t=1}^T \| \nabla f(\boldsymbol{w}_t) \|_2^2 \leq \frac{9 (F_1 L_{2,\infty} (\varsigma + \sigma) \tilde{A})^{2/3}}{(mnT)^{2/3}} + \frac{(72 F_1 L_{2,\infty} \sigma)^{2/3} + 64 F_1 L_{2,\infty} (2 + \tilde{A}/(mn))}{T}$$

$$= \tilde{O} \left( \frac{1}{(mnT)^{2/3}} + \frac{1}{T} \right).$$

We can also prove an accelerated rate for CD-GraB if we additionally assume the P.L. condition:

**Assumption 4** (**P.L. Condition**). *We say the loss function $f$ fulfills the P.L. condition if there exists $\mu > 0$ such that for any $\boldsymbol{w} \in \mathbb{R}^d$, $\frac{1}{2}\|\nabla f(\boldsymbol{w})\|_2^2 \geq \mu(f(\boldsymbol{w}) - \inf_{\boldsymbol{v} \in \mathbb{R}^d} f(\boldsymbol{v}))$.*

**Theorem 3.** *Suppose that Assumptions 1, 2, 3, and 4 hold. For any $\delta > 0$, we set constants $\tilde{W}$ and $C_3$ to be*

$$C_3 = \frac{(F_1 + \sigma^2/L_{2,\infty})\mu^2}{224 L_{2,\infty}^2 (\varsigma + \sigma)^2 \tilde{A}^2} \quad and \quad \tilde{W} = W_0(T^2 m^2 n^2 C_3),$$

*where $\tilde{A}$ comes from Theorem 1, $F_1$ is from Theorem 2, and $W_0$ is the Lambert-W function. If we set learning rate $\alpha = \frac{2\tilde{W}}{Tn\mu}$ and if the number of epochs $T$ satisfies*

$$T \geq 10 + \frac{1}{\mu} 32 L_{2,\infty}(2 + \tilde{A}/(mn))W_0((mnT)^2 C_3) = \tilde{O}(1),$$

*then, with probability at least $1 - T\delta$, it holds that*

$$F_{T+1} \leq \frac{1}{(mnT)^2}\left(\frac{(F_1 + L_{2,\infty}^2 \sigma^2)\tilde{W}}{C_3} + \frac{112 L_{2,\infty}^2 (\varsigma + \sigma)^2 \tilde{A}^2 \tilde{W}^2}{\mu^3}\right) = \tilde{O}\left(\frac{1}{(mnT)^2}\right),$$

*where $F_{T+1} = f(\boldsymbol{w}_{T+1}) - \inf_{\boldsymbol{w} \in \mathbb{R}^d} f(\boldsymbol{w})$.*

We prove Theorems 2 and 3 in the Appendix. Together, they show that CD-GraB exhibits a linear speedup in the number of workers $m$ over GraB [24]'s convergence rates ($\tilde{O}((nT)^{-2/3})$ and $\tilde{O}((nT)^{-2})$, respectively),[10] under both smoothness and the P.L. condition. Further, CD-GraB's convergence rate of $\tilde{O}((mnT)^{-2})$ is faster than many previous rates,[11] such as the high probability bound of $\tilde{O}((mn)^{-1}T^{-2})$ for D-RR in Yun et al. [48].

## 5 CD-GraB in Practice: Distributed and Simulation Experiments

We next verify CD-GraB's accelerated convergence on a variety of empirical tasks.[12] For ease of comparison, we follow the experimental plan from the original GraB paper,[13] and add some additional large-scale logistic regression experiments. We also run an ablation study to isolate the effects of different improvements in CD-GraB. We do this because online PairBalance exhibits performance benefits that are separate from parallelism — namely, removing the need for gradient centering with a stale mean and allowing for higher learning rates (Section 3.2).[14]

**Evaluating CD-GraB's convergence speedup.** We use the following three tasks for evaluating distributed training efficiency: logistic regression on a large-scale mortgage application (New York 2017 subset, 244,107 examples with 18 features) [7] (Figure 3a), Long Short-Term Memory (LSTM) [17] on the WikiText-2 dataset [29] (Figure 3b), and autoregressive Multi-Layer Perceptron (MLP) on the M4 Weekly dataset [25] (Figure 3c). We measure the loss incurred on the entire training set (Full Train Loss) and task-appropriate test metrics during evaluation, with respect to both the number of epochs and wall-clock time. Regarding test metrics, we measure test accuracy for the mortgage application, perplexity for WikiText-2, and SMAPE for M4. Additional details regarding the datasets, models, and test metrics can be found in the Appendix.

For all three tasks, we use a single 128 GiB memory machine with 4 NVIDIA GeForce RTX 2080 Ti GPUs. For the mortgage application and WikiText-2 (Figures 3a and 3b), we launch $m = 4$ workers (processes), where each worker runs on one GPU. For the M4 task, we launch $m = 32$ workers, where each of the 4 GPUs hosts 8 process workers. We use NCCL as the distributed communication backend [33] for the mortgage application and WikiText-2 tasks, and GLOO [1] as the distributed communication backend for the M4 task.

As shown in Figure 3, we compare CD-GraB's convergence to the standard distributed-training example-ordering method: random reshuffling (D-RR). From all subfigures in Figure 3, we observe that CD-GraB outperforms the D-RR baseline significantly and consistently: CD-GraB exhibits better training loss and test metrics, measured against both the number of epochs and wall-clock time. We also note that the

---

[10]For centralized GraB, the total number of examples $N = n$ and $m = 1$.

[11]These exclusively focus on the P.L. case, so we compare CD-GraB to them under the same condition.

[12]Our GitHub repository is https://github.com/GarlGuo/CD-GraB.

[13]Following Lu et al. [24], for our LSTM experiment on WikiText-2, we set the embedding dimension to 32. We note that we can improve perplexity if we set the dimension higher.

[14]GraB can also implement online PairBalance, in place of Balance [22] (Appendix).

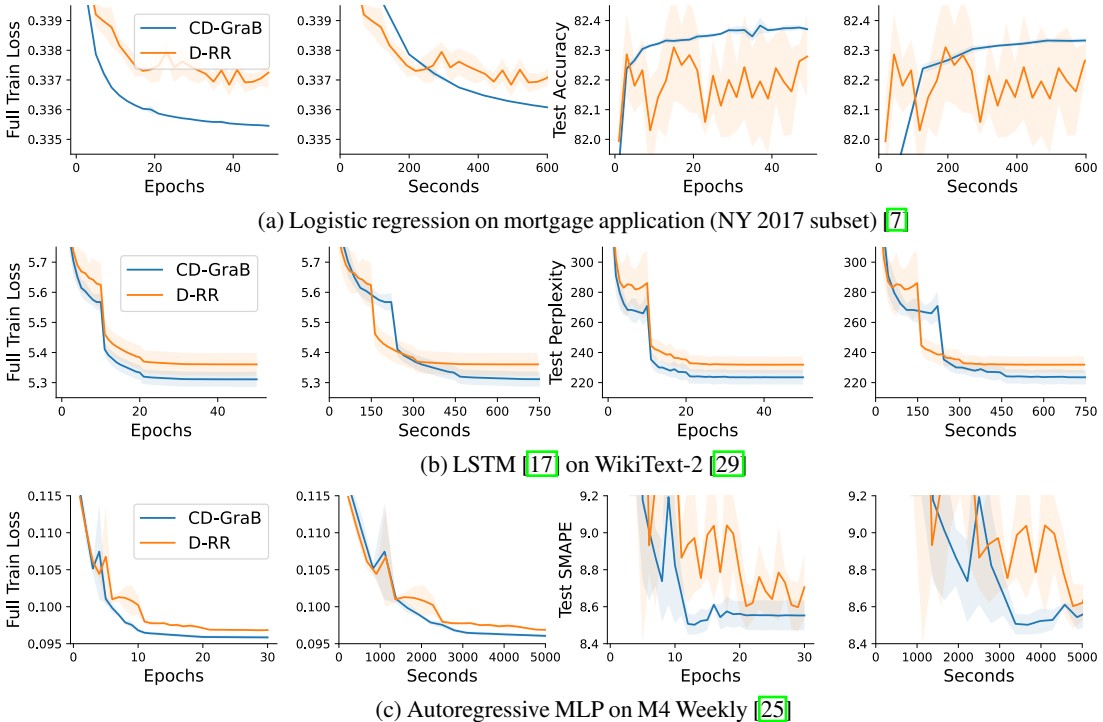

(a) Logistic regression on mortgage application (NY 2017 subset) [7]

(b) LSTM [17] on WikiText-2 [29]

(c) Autoregressive MLP on M4 Weekly [25]

Figure 3: Convergence of CD-GraB in comparison to D-RR. For each experiment, we show train loss over epochs and time (**left** of each subfigure) and test performance over epochs and time (**right** of each subfigure). We run at least 3 random seeds, and plot the mean ± STD.

results for CD-GraB are much smoother than for D-RR. This is likely due to the variance of stochastic gradients during training, which CD-GraB reduces as a side-effect (so, too, does GraB, in comparison to RR). For smoother D-RR results, we can reduce the learning rate (Appendix). CD-GraB allows for the use of a larger learning rate, which accelerates training while preserving the final model's performance.

**Ablation simulation study: the importance of coordination at large scale.** CD-GraB has several design benefits over the original centralized GraB algorithm [24]: coordinating parallel workers' specific permutations using PairBalance on the server (Algorithm 2) and removing the dependency on a stale mean (Section 2.1), which enables the ability to using larger learning rates reliably (Section 3.2). Clearly, not all of these benefits come directly from distributing training. For example, being able to use larger learning rates, is a side effect of our solution to develop CD-GraB, not our main contribution. Therefore, we run a simulation ablation study to disentangle the relative importance of each of CD-GraB's efficiency benefits over GraB. To do so, we compare the convergence of CD-GraB to two additional baselines in the distributed setting, beyond D-RR: (1) **ID-GraB (Bal)**, where each independent worker runs GraB locally using RandomizedBalance (subroutine in Algorithm 2) to perform gradient vector balancing; (2) **ID-GraB (PairBal)**, where each independent worker runs GraB locally using PairBalance.

Figure 4 summarizes the results, with convergence curves for $m \in \{4,8,16,32,64\}$ workers training LeNet on CIFAR-10. We choose this task and architecture to cohere with the experiments done in the original GraB paper. For these experiments, we denote $B$ to be the *aggregated* minibatch across all the workers, which refers to the number of stochastic examples used for an overall optimization step; each worker thus has a subset of this minibatch — an equivalently-sized subset of $B$ examples.[15] We make two main observations. First, when scaling up training with more workers, CD-GraB converges increasingly faster than the no-coordination-ordering methods **ID-GraB (Bal)** and **ID-GraB (PairBal)**. This result aligns with our theory and intuition that, when the number of workers $m$ increases, the parallel herding bound (8) will increase linearly if there is no coordination. Second, as we scale up to larger $m$, the convergence curves of **ID-GraB (Bal)** and **ID-GraB (PairBal)** gradually approach

---

[15]For example, if we have 4 workers with an aggregated minibatch size of 32, each worker would compute their respective local gradients with 8 examples, and then all-reduce these gradients to obtain the aggregated minibatch gradient for all 32 examples for the optimization step. We discard $N \bmod B$ examples at random to ensure $n$ examples per worker.

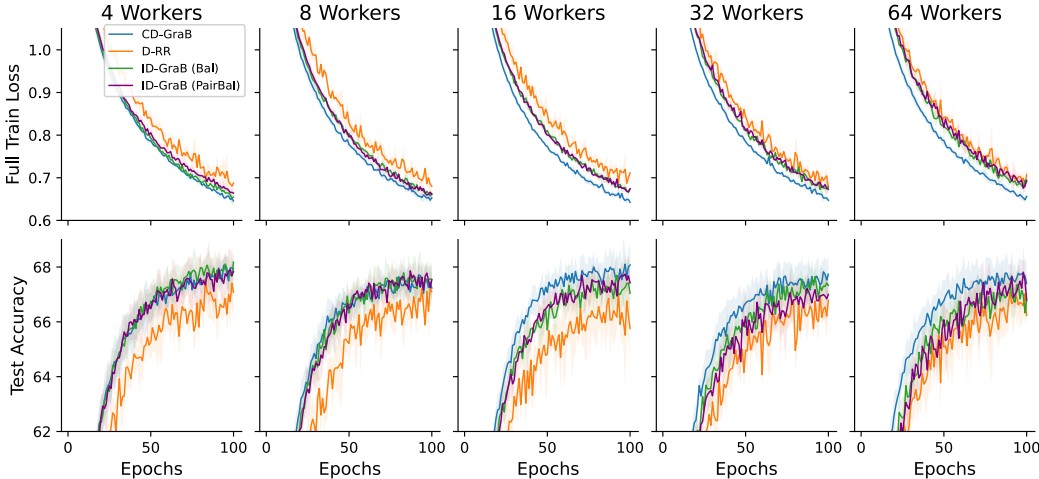

Figure 4: Convergence for CD-GraB, D-RR, ID-GraB (Bal), and ID-GraB (PairBal) training LeNet on CIFAR-10, with $m \in \{4,8,16,32,64\}$ workers. For each, the aggregated minibatch size per update is 64.

the curve for D-RR: at larger scales, herding-based example ordering will be no better than randomly permuting the dataset. Both observations give strong evidence that coordination (i.e., running online PairBalance on the server to coordinate per-worker permutations) is critical for accelerating training.

We note that all of these experiments use SGD, since both the theoretical results of the original GraB paper and our results for CD-GraB here are for SGD. In the Appendix, we additionally include results for training GPT-2 on WikiText-103, for which we use AdamW as the optimizer. We find that GraB with AdamW works in practice; however, our theory results do not directly apply to these experiments. We additionally include results on memory usage in the Appendix, which show that CD-GraB results in negligible overhead in practice.

## 6    Conclusion and Future Work: Toward an Order Server Architecture

We elevate the benefits of provably faster, permutation-based example ordering to the contemporary ML distributed-training setting. We focus on reformulating the online **Gra**dient **B**alancing algorithm (GraB) [24] because, even though it is the provably optimal permutation-based example-ordering method [6], it is limited by design to *centralized* settings (Section 3.1). To overcome these limitations, we redesign GraB's herding and balancing framework to account for parallel workers: A *parallel herding* objective, which we solve with an online PairBalance subroutine, based on key insights from kernel thinning [3, 10, 11]. PairBalance operates on ordered *pairs* of vectors to do *balancing*, which enables our full-stack, low-overhead, *Coordinated* and *Distributed* online CD-GraB algorithm. We give a full specification of our online CD-GraB algorithm (Section 3.3), provide convergence rate guarantees regarding its speedups on both 1) smooth non-convex and 2) P.L. objectives (Section 4), and verify these speedups in practice on single-node distributed tasks and a simulated ablation study (Section 5).

Both our theory and experiments demonstrate that CD-GraB really shines when there are multiple training epochs (Appendix). This is another reason that we do not emphasize experiments involving fine-tuning pre-trained models like GPT-2, as fine-tuning can be achieved in just a couple of epochs. As noted above, it is also more common to train such models using optimizers from the Adam family. In future work, we intend to extend the theory on GraB and CD-GraB to such optimizers, which would make the results on optimal, permutation-based example ordering more useful for base-model pre-training.

Pre-training from scratch would demonstrate the tremendous power of CD-GraB to scale to very large models; however, we did not have the training budget to perform such experiments for the present work. Further, to truly exercise the benefits of CD-GraB in such large-scale settings, future work should investigate moving beyond the single-node setup that we present. Notably, to train larger models, our results suggest a novel distributed training architecture. The ordering operation performed by the server (Algorithm 4) is *not* very latency sensitive; the server has the duration of the entire epoch $t$ to compute the new permutations for the next, $t+1$ epoch. Given this relaxed latency requirement, and the success of our algorithmic results, it would be an exciting direction for future ML-systems research to invest in building an *Order Server* architecture. Such an architecture, which could be composed with traditional parameter servers, would afford the scalability benefits of CD-GraB to a host of massive-scale ML applications.

## Acknowledgments

A. Feder Cooper is supported by Christopher De Sa's NSF CAREER grant. Yucheng Lu is supported by Meta Ph.D. Fellowship. We also acknowledge a gift from SambaNova Systems.

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
