# A    Glossary

| Term/Symbol | Explanation |
|---|---|
| GraB | Centralized online Gradient Balancing algorithm. We use this term to refer to the centralized algorithm developed in the original GraB paper (Lu et al. [24]). |
| CD-GraB | Coordinated and distributed online Gradient Balancing algorithm. We use this term to refer to the algorithm that constitutes our main contribution. |
| ID-GraB | Independent and distributed online gradient balancing. We implement this for our ablation study, to compare with coordinated and distributed GraB. There are two variants: One which uses the original GraB paper's online Balance algorithm (ID-GraB (Bal)), and one which implements our online PairBalance algorithm (ID-GraB (PairBal)). |
| RR | Random reshuffling algorithm. We use this to refer to its centralized variant. |
| D-RR | Distributed random reshuffling algorithm. |
| SO | Shuffle Once algorithm. |
| $\boldsymbol{x}$ | Data-example vector; we do not use this in the math in the main paper, but do refer to examples in our schematic description for PairBalance ordering in Figure 1. |
| $\boldsymbol{z}$ | Vector (for illustration under the herding context). For GraB and CD-GraB, these are gradients. |
| $\bar{\boldsymbol{z}}$ | The average vector (for illustration under the herding context). |
| $\boldsymbol{z}_j$ | The $j$-th component of a vector (for illustration under the herding context). |
| $\boldsymbol{z}_{i,j}$ | The $j$-th component of the gradient on worker $i$ (for illustration under our parallel herding framework). |
| $\boldsymbol{w}$ | Parameters / model-weights vector. |
| $f$ | Loss function. |
| $\nabla f(\boldsymbol{w})$ | Global loss gradient. |
| $\nabla f^i(\boldsymbol{w})$ | Local $i$-th worker's loss gradient. |
| $\nabla f^i(\boldsymbol{w};j)$ | Local $i$-th worker's, $j$-th example's loss gradient. |
| $\pi$ | A permutation; we study permutation-based example orderings. |
| $T$ | Number of epochs. |
| $t$ | Index for iterating over $T$ epochs. |
| $m$ | Number of workers (in this paper, workers are processes, potentially on different GPUs but on the same node). $m=1$ in the centralized setting. |
| $i$ | Index for iterating over $m$ workers . |
| $n$ | Number of training-data examples per worker; equivalent to $\frac{N}{m}$. |
| $N$ | Number of total training-data examples. $N=n$ in the centralized setting. |
| $j$ | Index for iterating over examples. |
| $\boldsymbol{g}$ | Gradient, taken with respect to the model weights $\boldsymbol{w}$ and data examples $\boldsymbol{x}$. |
| $\boldsymbol{g}_j^i$ | Gradient associated with the $j$-th data example $\boldsymbol{x}$ on worker $i$. |
| $s$ | A sign, either $+1$ or $-1$; related to the signed herding problem. |
| $s_j^i$ | A sign, either $+1$ or $-1$, computed according to the $j$-th example gradient $\boldsymbol{g}_j^i$ for worker $i$; to be associated with the example $\boldsymbol{x}_j$ when determining a permutation ordering using Algorithm 1. |

# B  Additional Details on the CD-GraB Algorithm and online PairBalance

In this Appendix, we provide more details on related work and our contributions. To start, we give a unified description of our online CD-GraB algorithm with prior work on herding, vector balancing, and kernel thinning (Appendix B.1), some more details on Alweiss et al. [2] that we elide in the main paper due to space constraints (Appendix B.2), conceptual details on implementing CD-GraB with a parameter server (Appendix B.3), and implementing our improved balancing algorithm (online PairBalance) in a centralized fashion to get additional improvements for GraB (Appendix B.4).

## B.1  Distinguishing our contributions

We summarize our contributions in relation to prior work in a concise format. This kind of presentation would not be easily understandable without the appropriate background and context that we provide in the paper. This is why present it here, in the Appendix, so that (ideally) this is seen by the reader after finishing the main paper.

We emphasize that it is prior work that:

- Formulates the herding objective and solves it with vector balancing [16, 45] (Algorithm 1).

- Leverages ideas from herding and vector balancing (above) in an optimization setting to do permutation-based example ordering [24].

- Observes and proves that it is possible to solve the herding objective in $\tilde{O}(1)$ by only examining differences on pairs of examples (the overarching idea of PairBalance [10], which relies on the online RandomizedBalance subroutine [2]; see Algorithm 2).

Our contributions are to bring together all of this prior work in a novel way. We

- Translate the herding and balancing framework to the parallel setting via defining a parallel herding objective (8).

- Leverage prior work on herding in an optimization setting [24] so that we can do parallel herding in an optimization setting (Section 3).

- Execute *online* pair balancing on a server (Algorithm 2 on a running sum, Figure 1), i.e., do pair balancing in a streaming and asynchronous (rather than blocking) fashion from gradient vectors produced on distributed workers (Algorithm 2), on the flattened sequenced of paired-difference gradients (Section 3.2); this leads to an improvement over GraB, which relies on a stale mean.

## B.2  More details on RandomizedBalance from Alweiss et al. [2]

In the subroutine for RandomizedBalance in Algorithm 2, we elide details about how the probability $p$ is computed exactly as in Alweiss et al. [2]. We provide a more complete specification in Algorithm 5 written in terms of a single input vector (which, for us, is the vector containing the difference between adjacent gradients). Note that the difference here is in the use of a required parameter, constant upper bound $w$, which is used to compute the probability $p$. For clarity of presentation in the subroutine in Algorithm 2, we have set $w = 1$. Alweiss et al. [2] sets this threshold differently, which we still elide for simplicity.

---

**Algorithm 5** Probabilistic Balancing with Logarithm Bound [Alweiss et al. [2]]

---

    **require:** parameter $w$, used to compute probability
    **input:** current running sum $r$ vector, vector $z_{\text{diff}}$
1: **if** $|\langle r, z_{\text{diff}} \rangle| > w$ or $\|r\|_\infty > w$ **then**
2:     **Fail**
3: **end if**
4: **compute:** $p \leftarrow \frac{1}{2} - \frac{\langle r, z_{\text{diff}} \rangle}{2w}$
5: **compute:** $s \leftarrow +1$   with probability   $p$;
               $s \leftarrow -1$   with probability   $1 - p$
6: **update:** $r \leftarrow r + s z_{\text{diff}}$
7: **return:** $s, r$

---

In practice, we actually do not use RandomizedBalance in our online PairBalance. We use the deterministic, greedy-ordering algorithm from the original Lu et al. [24, Algorithm 5] paper:

---

**Algorithm 6** Balancing without normalization [Lu et al. [24]]

---

    **input:** current running sum $r$ vector, vector $z_{\text{diff}}$
1: **if** $\|r + z_{\text{diff}}\| < \|r - z_{\text{diff}}\|$ **then** $s \leftarrow +1$ **else** $s \leftarrow -1$
2: **update:** $r \leftarrow r + s z_{\text{diff}}$
3: **return:** $s, r$

---

Note that, unlike Alweiss et al. [2] (Algorithm 5), Algorithm 6 from Lu et al. [24] cannot end up in a failure state.

In Alweiss et al. [2], Theorem 1.1 proves the $\tilde{O}(1)$ probabilistic bound for Algorithm 5 (See Theorem 1 for a restatement of this result in terms of our work). Corollary 7 of Dwivedi and Mackey [10] re-proves this result (which they mislabel as Alweiss et al. [2], Theorem 1.2, see Dwivedi and Mackey [10, Appendix R, p. 69]). They improve the constants and have a less conservative setting of the thresholds $w$. The proof is also very short and elegant, by relying on their Theorem 3.

## B.3 Implementing CD-GraB with a parameter server

For our implementation of CD-GraB, we use a parameter server architecture [20]. For our purposes, this just entails computing the average gradient (used to update the model on all workers) on the server side. That is, the server (other than determining the ordering for the next epoch) also has the function of aggregating gradient information (in this case, a simple mean) to send back to the workers.

We have the server compute the average $j$-th gradient for illustrative purposes. We could, instead, implement the computation the average gradient as an all-reduce operation, in which each worker broadcasts their gradients to all other workers, so that they can each locally compute the average gradient to update their local models. We implement CD-GraB using a parameter server pattern to show that this is a plausible architecture to use with our coordinated and distributed example ordering algorithm. We could also implement a full parameter server system, for which the server also coordinates global model updates.

If we kept everything in our implementation the same and switched to all-reduce, then we would no longer be following a parameter server paradigm. In this case, the server would just function to determine example orders. It is this kind of paradigm that suggests the abstraction of an *order server*, which we mention briefly in Section 6: A server whose sole responsibility is coordinating worker information to determine example ordering.

In future work, we intend to explore a host of architectural possibilities — of building a full system that incorporates both traditional parameter server aspects with our new abstraction of an order server. For example, we could have parameter servers and order servers work in tandem in a distributed system to perform model training. To move beyond the single-node implementation we present in this paper, we intend to investigate the benefits and trade-offs associated with such design decisions in an actual implemented system.

## B.4 Centralized online PairBalance

In Section 3.3, we provide a schematic diagram of how online PairBalance works for a distributed implementation using a parameter server (Figure 1). We also claim in Section 3 that online PairBalance can be applied to the original centralized GraB algorithm for improved empirical performance. We provide a schematic here, in Figure 5 (analogous to Figure 1), for online PairBalance for centralized GraB.

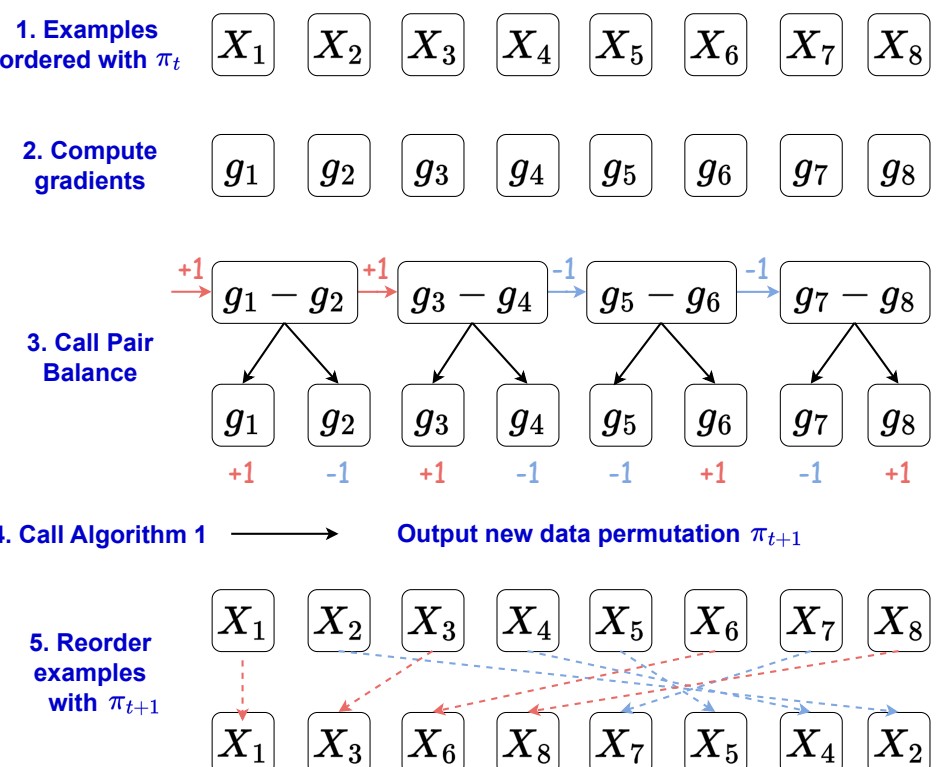

Figure 5: Schematic representation of online PairBalance for centralized GraB.

We also provide empirical results comparing GraB's Balance routine to the online PairBalance routine that we instead use in this work. We observe that both PairBalance and Balance would have similar convergence rates under centralized settings, and both outperform RR.

This experiment justifies the uses of PairBalance even in centralized learning settings. PairBalance theoretically tolerates higher learning rates and, as we will justify in Appendix D.1.2, is more memory-efficient than Balance. In short, PairBalance an excellent substitute for Balance when running GraB.

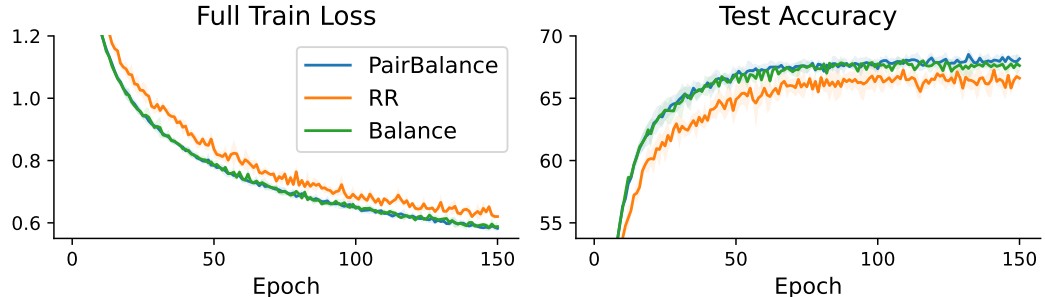

Figure 6: Convergence for centralized online PairBalance on LeNet on CIFAR-10. We use the identical set of hyperparameters ($\alpha$ = 1e-3, weight decay = 1e-2, momentum = 0.9, $B = 64$) as in the scaling experiments as in Figure 4.

# C   Proof Results

We present supporting results, which we use to prove the main results presented in Section 4. First, we show how to analyze the parallel herding bound in terms of a single step over the server-side PairBalance algorithm (Appendix C.1). We then include some additional observations/notation (Appendix C.2), which we use in the remaining intermediate results. We prove some intermediate results about how much the loss can change over the course of one epoch, assuming smoothness (Appendix C.3) and bounded gradient variance and heterogeneity (Appendix C.4). We combine these results to get one more intermediate result about the maximum the loss can change on average over many epochs (Appendix C.5), which we then use altogether to prove the two theorems that we present in the main paper (Appendix C.6).

## C.1   Analyzing the parallel herding bound

In the main paper, we cover how CD-GraB runs on both the worker- and server-side. In this section, we dive deeper into the example-ordering part of CD-GraB, and demonstrate in theory how server-side online PairBalance reduces the parallel herding bound (8), as formulated in Section 3. We conclude this section by presenting Lemma 1, which shows server-side PairBalance is able to iteratively reduce the parallel herding bound.

To begin, we formalize our illustration over a group of vectors (since vector balancing, including PairBalance, does not inherently involve an optimization context until we use it in our online setting on gradients). Without loss of generality, we assume that the $N$ examples are divided evenly among the $m$ workers and that $n$ is even. That is, we consider that we are given a set of vectors $z_{i,j} \in \mathbb{R}^d$ for $i \in [m]$ and $j \in [n]$ evenly located on $m$ workers (i.e., $n = \frac{N}{m}$), where $z_{i,j}$ denotes the $j$-th vector located on the $i$-th worker. Now denote $\pi_i$ as the original permutation of the vectors on worker $i$. Consider running Algorithm 7 on the server side over these $N$ vectors.

---

**Algorithm 7** Server-side PairBalance over a set of vectors (one step)

---

   **require:** $m$ workers, $n := \frac{N}{m}$ vectors per worker
   **input:** initial permutations for all the workers $\{\pi_i\}_{i=1}^m$
1: **initialize:** new permutations for all the workers $\{\pi_i'\}_{i=1}^m$
2: **initialize:** running partial sum $h = 0$
3: **initialize:** new indices front (left) pointer $\{l_i = 1\}_{i=1}^m$
4: **initialize:** new indices back (right) pointer $\{r_i = 1\}_{i=1}^m$
5: **for** example $j := 1...n$ **do**
6:     **for** worker $i := 1...m$ **do**
7:         **if** $j \bmod 2 = 0$ **then**      ▷ If at an even index, i.e., can examine a full pair of examples
8:             $h, s_{j-1}^i, s_j^i \leftarrow \mathsf{PairBalance}(h, z_{j-1}^i, z_j^i)$
9:             **if** $s_{j-1}^i = +1$ **then**
10:                 $\pi_i'(l_i) = j-1;\quad l_i = l_i + 1$    ▷ Append first in pair to the front/left
11:                 $\pi_i'(r_i) = j;\quad r_i = r_i - 1$    ▷ Append second in pair to the right/back
12:             **else**
13:                 $\pi_i'(l_i) = j;\quad l_i = l_i + 1$    ▷ Append second in pair to the left/front
14:                 $\pi_i'(r_i) = j-1;\quad r_i = r_i - 1$    ▷ Append first in pair to the right/back
15:             **end if**
16:         **end if**
17:     **end for**
18: **end for**
19: **output:** new permutations for all $m$ workers $\{\pi_i'\}_{i=1}^m$

---

Figure 7: One-step PairBalance algorithm on the server side to solve the parallel herding problem (8). This algorithm can be seen as a prototype for Algorithms 3 and 4, without the optimization context.

It follows, in the following Lemma 1, that we can get the parallel herding bound with the output permutations $\{\pi_i'\}_{i=1}^m$ from Algorithm 7:

**Lemma 1.** *Suppose that we have a set of vectors $z_{i,j} \in \mathbb{R}^d$ for all $i, i' \in [m]$ and for all $j, j' \in [n]$ that satisfies*

$$\left\| \sum_{i=1}^{m} \sum_{j=1}^{n} z_{i,j} \right\|_{\infty} \leq c_1 \qquad and \qquad \left\| z_{i',j'} - \frac{1}{mn} \sum_{i=1}^{m} \sum_{j=1}^{n} z_{i,j} \right\|_{\infty} \leq c_2$$

*for some constants $c_1 > 0$ and $c_2 > 0$. If we run Algorithm 7 over these vectors, then, for any $\delta > 0$, it holds with probability at least $1 - \delta$ that*

$$\max_{l \in [n]} \left\| \sum_{i=1}^{m} \sum_{j=1}^{l} z_{i,\pi_i'(j)} \right\|_{\infty} \leq \frac{1}{2} \max_{l \in [n]} \left\| \sum_{i=1}^{m} \sum_{j=1}^{l} z_{i,\pi_i(j)} \right\|_{\infty} + c_1 + \tilde{A} c_2,$$

*where $\tilde{A}$ comes from Theorem 1.*

Lemma 1 shows that PairBalance reduces the parallel herding objective (8) towards a constant (invariant to $n$) at each step. This implies that, if we repeatedly call PairBalance on a given permutation, it will return a permutation that guarantees the parallel herding bound to be $\tilde{O}(1)$.

*Proof.* We prove this lemma by defining the following auxiliary sequence of pair differences, as in Section 3.2

$$y_{n \cdot (k-1)+i} = z_{i,\pi_i(2k-1)} - z_{i,\pi_i(2k)}, \forall k \in [n/2],$$

which we also can refer to as $\{y_j\}_{j=1}^{mn/2}$.

We also leverage Theorem 1, which we reprint below for clarity of presentation:

**Theorem 1 (Corollary 7, Dwivedi and Mackey [10]).** *Consider any vectors $\{z_j\}_{j=1}^{N}$ ($z_j \in \mathbb{R}^d$) with $\|z_j\|_2 \leq 1$ supplied as input to the RandomizedBalance subroutine in Algorithm 2. Then for any $\delta > 0$, with probability at least $1 - \delta$, RandomizedBalance outputs a sequence of signs $\{s_j\}_{j=1}^{N} \in \{-1, 1\}$ that satisfy $\max_{k \in [N]} \left\| \sum_{j=1}^{k} s_j z_j \right\|_{\infty} \leq \tilde{A}$, where $\tilde{A} = \sqrt{2 \log\left(\frac{4d}{\delta}\right) \log\left(\frac{4N}{\delta}\right)} = \tilde{O}(1)$.*

Note that the reordering part of Algorithm 7 (line 8) gives a sequence of signs $\{s_j\}_{j=1}^{mn/2}$. Therefore, by Theorem 1, the sequence $\{y_j\}_{j=1}^{mn/2}$ satisfies

$$\max_{P \in [mn/2]} \left\| \sum_{p=1}^{P} s_p y_p \right\|_{\infty} \leq 2\tilde{A} c_2, \tag{9}$$

since (based on what is given in Lemma 1)

$$\left\| y_{n(k-1)+i} \right\|_{\infty} \leq \left\| z_{i,\pi_{t,i}(2k-1)} - \frac{1}{mn} \sum_{i=1}^{m} \sum_{j=1}^{n} z_{i,j} \right\|_{\infty} + \left\| z_{i,\pi_{t,i}(2k)} - \frac{1}{mn} \sum_{i=1}^{m} \sum_{j=1}^{n} z_{i,j} \right\|_{\infty} \leq 2c_2.$$

Note that, if $s_{i,k}$ is the sign associated with $y_{n(k-1)+i}$, then $z_{i,\pi_{t,i}(2k-1)}$ and $z_{i,\pi_{t,i}(2k)}$ will receive opposite signs $s_{i,k}$ and $-s_{i,k}$, respectively.

We denote $x_{i,k}^{+}$ to be the example that receives sign $s_{i,k} = +1$ and $x_{i,k}^{-}$ to be the example that receives sign $s_{i,k} = -1$.

That is, if $s_{i,k} = +1$, then $x_{i,k}^{+} = z_{i,\pi_i(2k-1)}$, otherwise, if $s_{i,k} = -1$, then $x_{i,k}^{+} = z_{i,\pi_i(2k)}$; and, $x_{i,k}^{-}$ is the other term of the pair $\{z_{i,\pi_i(2k-1)}, z_{i,\pi_i(2k)}\}$.

Now, for $K \in [\frac{n}{2}]$, let

$$\kappa_{i,K} = \sum_{k=1}^{K}(z_{i,\pi_i(2k-1)} + z_{i,\pi_i(2k)}) \quad \text{and}$$

$$\upsilon_{i,K} = \sum_{k=1}^{K}(s_{i,k} z_{i,\pi_i(2k-1)} - s_{i,k} z_{i,\pi_i(2k)}).$$

Then

$$\sum_{k=1}^{K} x_{i,k}^{+} = \frac{1}{2}(\kappa_{i,K} + \upsilon_{i,K}) \quad \text{and} \quad \sum_{k=1}^{K} x_{i,k}^{-} = \frac{1}{2}(\kappa_{i,K} - \upsilon_{i,K}).$$

Now, observe that

$$\sum_{i=1}^{m} \kappa_{i,K} = \sum_{j=1}^{2K}\sum_{i=1}^{m} z_{i,\pi_i(j)} \quad \text{and} \quad \sum_{i=1}^{m} \upsilon_{i,K} = \sum_{p=1}^{mK} s_p y_p.$$

Therefore,

$$\max_{K \in [n/2]} \left\| \sum_{k=1}^{K}\sum_{i=1}^{m} x_{i,k}^{+} \right\|_{\infty} \leq \frac{1}{2}\left( \max_{K \in [n/2]} \left\| \sum_{i=1}^{m} \kappa_{K,i} \right\|_{\infty} + \max_{K \in [n/2]} \left\| \sum_{i=1}^{m} \upsilon_{K,i} \right\|_{\infty} \right)$$

$$\leq \frac{1}{2} \max_{K \in [n/2]} \left\| \sum_{j=1}^{2K}\sum_{i=1}^{m} z_{i,j} \right\|_{\infty} + \tilde{A}c_2 \quad \text{By substituting above and (9)}$$

$$\leq \frac{1}{2} \max_{k \in [n]} \left\| \sum_{j=1}^{k}\sum_{i=1}^{m} z_{i,j} \right\|_{\infty} + \tilde{A}c_2.$$

And similarly,

$$\max_{K \in [n/2]} \left\| \sum_{k=1}^{K}\sum_{i=1}^{m} x_{i,k}^{-} \right\| \leq \frac{1}{2}\left( \max_{K \in [n/2]} \left\| \sum_{i=1}^{m} \kappa_{K,i} \right\|_{\infty} + \max_{K \in [n/2]} \left\| \sum_{i=1}^{m} \upsilon_{K,i} \right\|_{\infty} \right)$$

$$\leq \frac{1}{2} \max_{k \in [n]} \left\| \sum_{j=1}^{k}\sum_{i=1}^{m} z_{i,j} \right\|_{\infty} + \tilde{A}c_2.$$

Applying the new permutation $\pi_i'(j)$ on the vectors $z_{i,\pi_i(j)}$, we get for each $i \in [m]$ the permuted sequence

$$x_{i,1}^{+},...,x_{i,n/2}^{+}, x_{i,n/2}^{-},...,x_{i,1}^{-}.$$

Thus, we need to bound the herding objective of the sequence

$$\sum_{i=1}^{m} x_{i,1}^{+},..., \sum_{i=1}^{m} x_{i,n/2}^{+}, \sum_{i=1}^{m} x_{i,n/2}^{-},..., \sum_{i=1}^{m} x_{i,1}^{-}.$$

If the partial sums above peak at $t_0 \leq n/2$, then we can bound the parallel herding objective as

$$\left\| \sum_{k=1}^{t_0}\sum_{i=1}^{m} x_{i,k}^{+} \right\|_{\infty} = \max_{K \in [n/2]} \left\| \sum_{k=1}^{K}\sum_{i=1}^{m} x_{i,k}^{+} \right\|_{\infty} \leq \frac{1}{2} \max_{k \in [n]} \left\| \sum_{j=1}^{k}\sum_{i=1}^{m} z_{i,j} \right\|_{\infty} + \tilde{A}c_2;$$

otherwise, we can bound the parallel herding objective as

$$\left\|\sum_{j=1}^{n}\sum_{i=1}^{m}\boldsymbol{z}_{i,j}-\sum_{k=1}^{m-t_0}\sum_{i=1}^{m}\boldsymbol{x}_{i,k}^{-}\right\|_{\infty} \leq \left\|\sum_{j=1}^{n}\sum_{i=1}^{m}\boldsymbol{z}_{i,j}\right\|_{\infty} + \left\|\sum_{k=1}^{m-t_0}\sum_{i=1}^{m}\boldsymbol{x}_{i,k}^{-}\right\|_{\infty}$$

$$\leq c_1 + \frac{1}{2}\max_{t\in[n]}\left\|\sum_{j=1}^{t}\sum_{i=1}^{m}\boldsymbol{z}_{i,j}\right\|_{\infty} + \tilde{A}c_2,$$

since in Algorithm 1 the list of vectors with negative signs is reversed before concatenated.

The claim follows. □

## C.2   Notation and observations

We begin with three notes that we will use throughout the intermediate results we present in this section. We will use the lemmas presented here to prove our main results: Theorems 2 and 3 in Appendix C.6.

1. **A single $t$-th update.** First, recall that one $t$-th step of the parameter update can be written as
$$\boldsymbol{w}_t^{j+1} = \boldsymbol{w}_t^j - \frac{\alpha}{m}\sum_{i=1}^{m}\nabla f^i(\boldsymbol{w}_t^j;\pi_{t,i}(j)), \quad \forall j\in[n]$$
We will use the convention $\boldsymbol{w}_{t+1} \triangleq \boldsymbol{w}_{t+1}^1 \triangleq \boldsymbol{w}_t^{n+1}$.

2. **The maximum amount a parameter can change over an epoch.** The key quantity in our proof is $\Delta_t$, which is the maximum amount that a parameter in $\boldsymbol{w}$ can change in epoch $t$. That is,
$$\Delta_t \triangleq \max_{k\in[n]}\left\|\boldsymbol{w}_t^{k+1} - \boldsymbol{w}_t\right\|_{\infty}$$
$$= \frac{\alpha}{m}\max_{k\in[n]}\left\|\sum_{j=1}^{k}\sum_{i=1}^{m}\nabla f^i(\boldsymbol{w}_t^j;\pi_{t,i}(j))\right\|_{\infty}. \tag{10}$$
Following this definition of $\Delta_t$, we note that the maximum amount that a parameter in $\boldsymbol{w}$ can change over two different epochs is $2\Delta_t$. That is, we observe
$$\left\|\boldsymbol{w}_t^j - \boldsymbol{w}_t^k\right\|_{\infty} \leq \left\|\boldsymbol{w}_t^j - \boldsymbol{w}_t\right\|_{\infty} + \left\|\boldsymbol{w}_t^k - \boldsymbol{w}_t\right\|_{\infty} \leq 2\Delta_t$$
$$\left\|\boldsymbol{w}_{t+1}^j - \boldsymbol{w}_t^k\right\|_{\infty} \leq \left\|\boldsymbol{w}_{t+1}^j - \boldsymbol{w}_{t+1}\right\|_{\infty} + \left\|\boldsymbol{w}_{t+1} - \boldsymbol{w}_t\right\|_{\infty} + \left\|\boldsymbol{w}_t^k - \boldsymbol{w}_t\right\|_{\infty} \leq \Delta_{t+1} + 2\Delta_t, \quad \forall j,k\in[n].$$
We make repeated use of this relation in the results that follow, which we typically will use in combination with the Lipschitz assumption to bound gradients of the same loss function but with different parameters.

3. **Bounding loss at epoch $t$.** We will denote $F_t = f(\boldsymbol{w}_t) - f(\boldsymbol{w}^*)$ where $\boldsymbol{w}^*$ is the minimizer of $f$ which we assume to be bounded from below.

## C.3   Assuming $L_{2,\infty}$-smoothness: results on the amount the loss can change over one epoch)

We will next prove an intermediate result regarding that bounds the loss $f$ at epoch $t+1$ in relation to the loss at the prior epoch $t$ (Lemma 2). That is, we prove results about how much the loss with respect to the parameters can change over the course of one epoch.

**Lemma 2.** *If the loss $f$ is $L_{2,\infty}$-smooth and the learning rate $\alpha \leq \frac{1}{nL_{2,\infty}}$, then*
$$f(\boldsymbol{w}_{t+1}) \leq f(\boldsymbol{w}_t) + \frac{\alpha n L_{2,\infty}^2}{2}\Delta_t^2 - \frac{\alpha n}{2}\|\nabla f(\boldsymbol{w}_t)\|_2^2. \tag{11}$$

*Proof.* We begin with the definition of $L_{2,\infty}$-smoothness, with respect to loss $f$:
$$f(\boldsymbol{w}_{t+1}) \leq f(\boldsymbol{w}_t) + \nabla f(\boldsymbol{w}_t)^\top(\boldsymbol{w}_{t+1} - \boldsymbol{w}_t) + \frac{L_{2,\infty}}{2}\|\boldsymbol{w}_{t+1} - \boldsymbol{w}_t\|_2^2$$

Also observe that

$$-\nabla f(\boldsymbol{w}_t)^\top(\boldsymbol{w}_t - \boldsymbol{w}_{t+1}) = -\frac{\alpha n}{2} 2\nabla f(\boldsymbol{w}_t)^\top\left(\frac{\boldsymbol{w}_t - \boldsymbol{w}_{t+1}}{\alpha n}\right)$$

$$= \frac{\alpha n}{2}\left(\left\|\nabla f(\boldsymbol{w}_t) - \frac{(\boldsymbol{w}_t - \boldsymbol{w}_{t+1})}{\alpha n}\right\|_2^2 - \|\nabla f(\boldsymbol{w}_t)\|_2^2 - \left\|\frac{(\boldsymbol{w}_t - \boldsymbol{w}_{t+1})}{\alpha n}\right\|_2^2\right). \tag{12}$$

Combining the above — i.e., the definition of $L_{2,\infty}$-smoothness with (12) — we get

$$f(\boldsymbol{w}_{t+1}) \le f(\boldsymbol{w}_t) + \frac{\alpha n}{2}\left\|\nabla f(\boldsymbol{w}_t) - \frac{(\boldsymbol{w}_t - \boldsymbol{w}_{t+1})}{\alpha n}\right\|_2^2 - \frac{\alpha n}{2}\|\nabla f(\boldsymbol{w}_t)\|_2^2 \qquad + \frac{\alpha n L_{2,\infty} - 1}{2\alpha n}\|\boldsymbol{w}_t - \boldsymbol{w}_{t+1}\|_2^2.$$

The last term on the right-hand side is $\le 0$ by the assumption that the learning rate $\alpha \le \frac{1}{nL_{2,\infty}}$. Therefore,

$$f(\boldsymbol{w}_{t+1}) \le f(\boldsymbol{w}_t) + \frac{\alpha n}{2}\left\|\nabla f(\boldsymbol{w}_t) - \frac{(\boldsymbol{w}_t - \boldsymbol{w}_{t+1})}{\alpha n}\right\|_2^2 - \frac{\alpha n}{2}\|\nabla f(\boldsymbol{w}_t)\|_2^2. \tag{13}$$

We next bound the second term on the right-hand side by $\Delta_t$ (10):

$$\left\|\nabla f(\boldsymbol{w}_t) - \frac{(\boldsymbol{w}_t - \boldsymbol{w}_{t+1})}{\alpha n}\right\|_2^2 = \left\|\frac{1}{mn}\sum_{j=1}^m\sum_{i=1}^n \nabla f^i(\boldsymbol{w}_t, \pi_t(j)) - \frac{1}{mn}\sum_{j=1}^m\sum_{i=1}^n \nabla f^i(\boldsymbol{w}_t^j; \pi_t(j))\right\|_2^2$$

$$\le \frac{1}{mn}\sum_{j=1}^m\sum_{i=1}^n\left\|\nabla f^i(\boldsymbol{w}_t, \pi_t(j)) - \nabla f^i(\boldsymbol{w}_t^j; \pi_t(j))\right\|_2^2$$

$$\le \frac{L_{2,\infty}^2}{mn}\sum_{j=1}^m\sum_{i=1}^n\left\|\boldsymbol{w}_t^j - \boldsymbol{w}_t\right\|_\infty^2,$$

where we have used $L_{2,\infty}$-smoothness (Assumption 3) in the last inequality. Substituting $\Delta_t$, we get

$$\frac{L_{2,\infty}^2}{mn}\sum_{j=1}^m\sum_{i=1}^n\left\|\boldsymbol{w}_t^j - \boldsymbol{w}_t\right\|_\infty^2 \quad \le \quad L_{2,\infty}^2\Delta_t^2.$$

Plugging the above into (13), we get

$$f(\boldsymbol{w}_{t+1}) \le f(\boldsymbol{w}_t) + \frac{\alpha n L_{2,\infty}^2}{2}\Delta_t^2 - \frac{\alpha n}{2}\|\nabla f(\boldsymbol{w}_t)\|_2^2,$$

yielding the claim. $\qquad\qquad\square$

We next build slightly on Lemma 2 to make two additional observations. First:

**Lemma 3.** *If the loss $f$ is $L_{2,\infty}$-smooth and the learning rate $\alpha \le \frac{1}{nL_{2,\infty}}$, then*

$$\frac{1}{T}\sum_{t=1}^T\|\nabla f(\boldsymbol{w}_t)\|_2^2 \le \frac{2F_1}{\alpha nT} + \frac{L_{2,\infty}^2}{T}\sum_{t=1}^T\Delta_t^2,$$

*where $F_1$ comes from Theorem 2.*

*Proof.* Using Lemma 2 and Jensen's inequality, we average (11) over $t \in [T]$ and match terms, yielding

$$\frac{1}{T}\sum_{t=1}^T\|\nabla f(\boldsymbol{w}_t)\|_2^2 \le \frac{2(f(\boldsymbol{w}_1) - f(\boldsymbol{w}_{T+1}))}{\alpha nT} + \frac{L_{2,\infty}^2}{T}\sum_{t=1}^T\Delta_t^2.$$

Substituting $F_1$, we get

$$\le \frac{2F_1}{\alpha nT} + \frac{L_{2,\infty}^2}{T}\sum_{t=1}^T\Delta_t^2,$$

yielding the claim. $\qquad\qquad\square$

We next build on Lemma 2 by further assuming the P.L. assumption holds.

**Lemma 4.** *If the loss $f$ is $L_{2,\infty}$-smooth, the learning rate $\alpha \leq \frac{1}{nL_{2,\infty}}$, and the P.L. assumption (Assumption 4) holds, then, for $\rho = 1 - \frac{\alpha n \mu}{2}$*

$$F_{T+1} \leq \rho^T F_1 + \frac{\alpha n L_{2,\infty}^2}{2} \sum_{t=1}^{T} \rho^{T-t} \left( \Delta_t^2 - \frac{1}{2L_{2,\infty}^2} \|\nabla f(\boldsymbol{w}_t)\|_2^2 \right).$$

*Proof.* From Lemma 2, we got (11), i.e.,

$$f(\boldsymbol{w}_{t+1}) \leq f(\boldsymbol{w}_t) + \frac{\alpha n L_{2,\infty}^2}{2} \Delta_t^2 - \frac{\alpha n}{2} \|\nabla f(\boldsymbol{w}_t)\|_2^2,$$

Applying the P.L. assumption (Assumption 4) to (11), we get

$$f(\boldsymbol{w}_{t+1}) \leq f(\boldsymbol{w}_t) + \frac{\alpha n L_{2,\infty}^2}{2} \Delta_t^2 - \frac{\alpha n}{4} \|\nabla f(\boldsymbol{w}_t)\|_2^2 - \frac{\alpha n}{4} \|\nabla f(\boldsymbol{w}_t)\|_2^2$$

$$\leq f(\boldsymbol{w}_t) + \frac{\alpha n L_{2,\infty}^2}{2} \Delta_t^2 - \frac{\alpha n \mu}{2}(f(\boldsymbol{w}_t) - f(\boldsymbol{w}^*)) - \frac{\alpha n}{4} \|\nabla f(\boldsymbol{w}_t)\|_2^2.$$

Subtracting $f^*$ from both sides, we get

$$f(\boldsymbol{w}_{t+1}) - f^* \leq \left( 1 - \frac{\alpha n \mu}{2} \right)(f(\boldsymbol{w}_t) - f^*) + \frac{\alpha n}{2} \left( L_{2,\infty}^2 \Delta_t^2 - \frac{1}{2} \|\nabla f(\boldsymbol{w}_t)\|_2^2 \right).$$

For $\rho = 1 - \frac{\alpha n \mu}{2}$, we then apply the above inequality recursively for $t \in [T]$, yielding the claim:

$$F_{T+1} \leq \rho^T F_1 + \frac{\alpha n L_{2,\infty}^2}{2} \sum_{t=1}^{T} \rho^{T-t} \left( \Delta_t^2 - \frac{1}{2L_{2,\infty}^2} \|\nabla f(\boldsymbol{w}_t)\|_2^2 \right).$$

$\square$

### C.4 Assuming bounded gradient variance and heterogeneity: results applying Algorithm 7

We next prove a result that builds on Lemma 1 and our one-step version of the server-side PairBalance algorithm (Algorithm 7).

We begin by introducing some additional notation. Namely, we will call $\pi^{-1}$ the operation that, given an example, yields the index in the permutation for that example. For instance, $\pi_{t+1,i}(j)$ returns the example at the $j$-th index for the $i$-th worker's $t+1$ permutation. Let us denote that example $\tau$. Then, $\pi_{t,i}^{-1} \pi_{t+1,i}(j)$ is equivalent to applying $\pi_{t,i}^{-1}$ to $\tau$: it takes the example $\tau$ and returns $\tau$'s associated index in the $i$-th worker's epoch $t$'s permutation (in this case, the prior epoch's permutation).

We will make use of this notation in the following Lemma.

**Lemma 5.** *Assume bounded gradient variance (Assumption 1), bounded gradient heterogeneity (Assumption 2), and $L_{2,\infty}$-smoothness (Assumption 3). For $t \in [T]$ and $\delta > 0$, if we apply Algorithm 7 to the gradients $\nabla f^i(\boldsymbol{w}_t^j; \pi_t^i(j))$ at epoch $t$ to produce the next permutation $\pi_{t+1,i}$ for epoch $t+1$, then, with probability at least $1 - \delta$,*

$$\Delta_{t+1} \leq \frac{1}{2}\Delta_t + \alpha L_{2,\infty}\left(4n + \frac{2\tilde{A}}{m}\right)\Delta_t + \alpha n L_{2,\infty}\Delta_{t+1} + \frac{\alpha(\varsigma + \sigma)\tilde{A}}{m} + \alpha n \|\nabla f(\boldsymbol{w}_{t+1})\|_2,$$

*where $\tilde{A}$ comes from Theorem 1.*

*Proof.* We start with the triangle inequality:

$$\left\| \sum_{j=1}^{k}\sum_{i=1}^{m} \nabla f^i(\boldsymbol{w}_{t+1}^j; \pi_{t+1,i}(j)) \right\|_\infty \leq \left\| \sum_{j=1}^{k}\sum_{i=1}^{m} \nabla f^i(\boldsymbol{w}_t^{\pi_{t,i}^{-1}\pi_{t+1,i}(j)}; \pi_{t+1,i}(j)) \right\|_\infty +$$

$$\left\| \sum_{j=1}^{k}\sum_{i=1}^{m} \left( \nabla f^i(\boldsymbol{w}_{t+1,i}^j, \pi_{t+1,i}(j)) - \nabla f^i(\boldsymbol{w}_{t,i}^{\pi_{t,i}^{-1}\pi_{t+1,i}(j)}; \pi_{t+1,i}(j)) \right) \right\|_\infty \qquad (14)$$

We use Lemma 1 to bound the first term on the right-hand side of (14) from Lemma 2.

That is, let
$$\boldsymbol{z}_{i,j} = \nabla f^i(\boldsymbol{w}_t^{\pi_{t,i}^{-1}(j)};j),$$
so that
$$\boldsymbol{z}_{i,\pi_{t+1,i}(j)} = \nabla f^i(\boldsymbol{w}_t^{\pi_{t,i}^{-1}\pi_{t+1,i}(j)};\pi_{t+1,i}(j)).$$

The upper bounds for $\left\|\boldsymbol{z}_{i,j} - \frac{1}{mn}\sum_{r,s}\boldsymbol{z}_{r,s}\right\|_\infty$ and $\left\|\sum_{i,j}\boldsymbol{z}_{i,j}\right\|_\infty$ are:

$$\left\|\nabla f^i(\boldsymbol{w}_t^j;\pi_{t,i}(j)) - \frac{1}{mn}\sum_{r=1}^m\sum_{s=1}^n\nabla f^s(\boldsymbol{w}_t^r;\pi_{t,s}(r))\right\|_\infty,$$

which are

$$\leq \left\|\nabla f^i(\boldsymbol{w}_t^j;\pi_{t,i}(j)) - \frac{1}{mn}\sum_{r=1}^m\sum_{s=1}^n\nabla f^s(\boldsymbol{w}_t^j;\pi_{t,s}(r))\right\|_\infty +$$

$$\left\|\frac{1}{mn}\sum_{r=1}^m\sum_{s=1}^n\nabla f^s(\boldsymbol{w}_t^j;\pi_{t,s}(r)) - \frac{1}{mn}\sum_{r=1}^m\sum_{s=1}^n\nabla f^s(\boldsymbol{w}_t^r;\pi_{t,s}(r))\right\|_\infty.$$

We can rewrite the above to be

$$\leq \left\|\nabla f^i(\boldsymbol{w}_t^j;\pi_{t,i}(j)) - \nabla f(\boldsymbol{w}_t^j)\right\|_\infty + \frac{L_{2,\infty}}{mn}\sum_{r=1}^m m\sum_{s=1}^n\left\|\boldsymbol{w}_t^j - \boldsymbol{w}_t^r\right\|_\infty$$

$$\leq \varsigma + \sigma + 2L_{2,\infty}\Delta_t,$$

by Assumptions 1, 2, and 3, and by the definition of $\Delta_t$ (10).

Now, observe that

$$\left\|\sum_{i=1}^m\sum_{j=1}^n\nabla f^i(\boldsymbol{w}_t^j;\pi_{t,i}(j))\right\|_\infty \leq \left\|\sum_{i=1}^m\sum_{j=1}^n\nabla f^i(\boldsymbol{w}_t^j;\pi_{t,i}(j)) - \sum_{i=1}^m\sum_{j=1}^n\nabla f^i(\boldsymbol{w}_{t+1};\pi_{t,i}(j))\right\|_\infty +$$

$$\left\|\sum_{i=1}^m\sum_{j=1}^n\nabla f^i(\boldsymbol{w}_{t+1};\pi_{t,i}(j))\right\|_\infty.$$

By using the above, we can rewrite the right-hand side to be

$$\leq \sum_{i=1}^m\sum_{j=1}^n L_{2,\infty}\left\|\boldsymbol{w}_t^j - \boldsymbol{w}_{t+1}\right\|_\infty + mn\|\nabla f(\boldsymbol{w}_{t+1})\|_\infty$$

$$\leq 2mnL_{2,\infty}\Delta_t + mn\|\nabla f(\boldsymbol{w}_{t+1})\|_2.$$

Therefore, by Lemma 1,

$$\max_{k\in[n]}\left\|\sum_{j=1}^k\sum_{i=1}^m\nabla f^i(\boldsymbol{w}_t^{\pi_{t,i}^{-1}\pi_{t+1,i}(j)},\pi_{t+1,i}(j))\right\|_\infty \leq \max_{k\in[n]}\left\|\sum_{j=1}^k\sum_{i=1}^m\nabla f^i(\boldsymbol{w}_t^j,\pi_{t,i}(j))\right\|_\infty$$

$$+ 2mnL_{2,\infty}\Delta_t + \|\nabla f(\boldsymbol{w}_{t+1})\|_2 + (\varsigma + \sigma + 2L_{2,\infty}\Delta_t)\tilde{A}.$$

The second term of the triangle inequality (14) can be bounded as

$$\left\|\sum_{j=1}^k\sum_{i=1}^m\left(\nabla f^i(\boldsymbol{w}_{t+1,i}^j;\pi_{t+1,i}(j)) - \nabla f^i(\boldsymbol{w}_{t,i}^{\pi_{t,i}^{-1}\pi_{t+1,i}(j)};\pi_{t+1,i}(j))\right)\right\|_\infty,$$

which is

$$\leq \sum_{j=1}^{k}\sum_{i=1}^{m}\left\|\boldsymbol{w}_{t+1,i}^{j}-\boldsymbol{w}_{t,i}^{\pi_{t,i}^{-1}\pi_{t+1,i}(j)}\right\|_{\infty}$$

$$\leq mnL_{2,\infty}(\Delta_{t+1}+2\Delta_{t}).$$

Substituting these bounds into the right-hand side of the triangle inequality (14), taking the max of both sides, and grouping terms, we get

$$\max_{k\in[n]}\left\|\sum_{j=1}^{k}\sum_{i=1}^{m}\nabla f^{i}(\boldsymbol{w}_{t+1,i}^{j},\pi_{t+1,i}(j))\right\|_{\infty} \leq \frac{1}{2}\max_{k\in[n]}\left\|\sum_{j=1}^{k}\sum_{i=1}^{m}\nabla f^{i}(\boldsymbol{w}_{t}^{j},\pi_{t,i}(j))\right\|_{\infty}$$
$$+L_{2,\infty}(4mn+2\tilde{A})\Delta_{t}+mnL_{2,\infty}\Delta_{t+1}+(\varsigma+\sigma)\tilde{A}$$
$$+mn\|\nabla f(\boldsymbol{w}_{t+1})\|_{2}.$$

Multiplying both sides by $\frac{\alpha}{m}$ and using the definition of $\Delta_{t}$ (10), we get the claim. $\qquad\square$

## C.5  Combining the prior intermediate results: proofs over multiple steps

**Lemma 6.** *If the learning rate $\alpha\leq\frac{1}{16L_{2,\infty}(2n+\tilde{A}/m)}$, then*

$$\frac{1}{T}\sum_{t=1}^{T}\Delta_{t}^{2}\leq\frac{21\alpha^{2}(\varsigma+\sigma)^{2}\tilde{A}^{2}}{m^{2}}+\frac{9\alpha^{2}n^{2}\sigma^{2}}{T}+21\alpha^{2}n^{2}\frac{1}{T}\sum_{t=1}^{T}\|\nabla f(\boldsymbol{w}_{t})\|_{2}^{2}.$$

*Proof.* First, we bound $\Delta_{1}^{2}$.

We start with a series of triangle inequalities:

$$\frac{\alpha}{m}\left\|\sum_{j=1}^{k}\sum_{i=1}^{m}\nabla f^{i}(\boldsymbol{w}_{1}^{j},\pi_{1,i}(j))\right\|_{\infty} \leq \frac{\alpha}{m}\left\|\sum_{j=1}^{k}\sum_{i=1}^{m}\nabla f^{i}(\boldsymbol{w}_{1}^{j},\pi_{1,i}(j))-\sum_{j=1}^{k}\sum_{i=1}^{m}\nabla f^{i}(\boldsymbol{w}_{1},\pi_{1,i}(j))\right\|_{\infty}$$
$$+\frac{\alpha}{m}\left\|\sum_{j=1}^{k}\sum_{i=1}^{m}(\nabla f^{i}(\boldsymbol{w}_{1},\pi_{1,i}(j))-\nabla f^{i}(\boldsymbol{w}_{1}))\right\|_{\infty}+\alpha k\|\nabla f(\boldsymbol{w}_{1})\|_{\infty}$$
$$\leq \frac{\alpha}{m}\sum_{j=1}^{k}\sum_{i=1}^{m}L_{2,\infty}\left\|\boldsymbol{w}_{1}^{j}-\boldsymbol{w}_{1}\right\|_{\infty}+\alpha k\sigma+\alpha k\|\nabla f(\boldsymbol{w}_{1})\|_{2}.$$

We next take the max of both sides with respect to $k\in[n]$:

$$\Delta_{1}\leq\alpha nL_{2,\infty}\Delta_{1}+\alpha n\sigma+\alpha n\|\nabla f(\boldsymbol{w}_{1})\|_{2}$$
$$\leq(1/32)\Delta_{1}+\alpha n\sigma+\alpha n\|\nabla f(\boldsymbol{w}_{1})\|_{2}\qquad\qquad\text{(since }\alpha\leq\frac{1}{32nL_{2,\infty}}\text{)}$$
$$\leq(32/31)\alpha n\sigma+(32/31)\alpha n\|\nabla f(\boldsymbol{w}_{1})\|_{2},$$

Squaring both sides:

$$\Delta_{1}^{2}\leq 3\alpha^{2}n^{2}\sigma^{2}+3\alpha^{2}n^{2}\|\nabla f(\boldsymbol{w}_{1})\|_{2}^{2}.\qquad\qquad(15)$$

Now, we use Lemma 5 to get the relationship between $\Delta_{t+1}$ and $\Delta_{t}$ for $t\in[T]$.

Recall that

$$\Delta_{t+1}\leq\frac{1}{2}\Delta_{t}+\alpha L_{2,\infty}\left(4n+\frac{2\tilde{A}}{m}\right)\Delta_{t}+\alpha nL_{2,\infty}\Delta_{t+1}+\frac{\alpha(\varsigma+\sigma)\tilde{A}}{m}+\alpha n\|\nabla f(\boldsymbol{w}_{t+1})\|_{2}$$

Because $\alpha \le \frac{1}{16L_{2,\infty}(2n+\tilde{A}/m)}$, we can rewrite the above as

$$\Delta_{t+1} \le \frac{1}{2}\Delta_t + (1/8)\Delta_t + (1/32)\Delta_{t+1} + \frac{\alpha(\varsigma+\sigma)\tilde{A}}{m} + \alpha n \|\nabla f(\boldsymbol{w}_{t+1})\|_2.$$

Squaring both sides:

$$(31/32)^2 \Delta_{t+1}^2 \le \frac{1}{2}\Delta_t^2 + 2\left((1/8)\Delta_t + \frac{\alpha(\varsigma+\sigma)\tilde{A}}{m} + \alpha n \|\nabla f(\boldsymbol{w}_{t+1})\|_2\right)^2$$

$$\le \frac{1}{2}\Delta_t^2 + (6/8^2)\Delta_t^2 + \frac{6\alpha^2(\varsigma+\sigma)^2\tilde{A}^2}{m^2} + 6\alpha^2 n^2 \|\nabla f(\boldsymbol{w}_{t+1})\|_2^2,$$

so that

$$\Delta_{t+1}^2 \le (32/31)^2(1/2+6/8^2)\Delta_t^2 + \frac{(32/31)^2 6\alpha^2(\varsigma+\sigma)^2\tilde{A}^2}{m^2} + (32/31)^2 6\alpha^2 n^2 \|\nabla f(\boldsymbol{w}_{t+1})\|_2^2$$

$$\le (2/3)\Delta_t^2 + \frac{7\alpha^2(\varsigma+\sigma)^2\tilde{A}^2}{m^2} + 7\alpha^2 n^2 \|\nabla f(\boldsymbol{w}_{t+1})\|_2^2. \tag{16}$$

We next sum (16) over $t \in [T-1]$ and add (15):

$$\Delta_1^2 + \sum_{t=2}^{T}\Delta_t^2 \le (2/3)\sum_{t=2}^{T}\Delta_{t-1}^2 + \frac{(T-1)7\alpha^2(\varsigma+\sigma)^2\tilde{A}^2}{m^2} + 3\alpha^2 n^2\sigma^2 + 7\alpha^2 n^2 \sum_{t=1}^{T}\|\nabla f(\boldsymbol{w}_t)\|_2^2$$

$$\frac{1}{T}\sum_{t=1}^{T}\Delta_t^2 \le (2/3)\frac{1}{T}\sum_{t=1}^{T}\Delta_t^2 + \frac{7\alpha^2(\varsigma+\sigma)^2\tilde{A}^2}{m^2} + \frac{3\alpha^2 n^2\sigma^2}{T} + 7\alpha^2 n^2\frac{1}{T}\sum_{t=1}^{T}\|\nabla f(\boldsymbol{w}_t)\|_2^2$$

$$\le \frac{21\alpha^2(\varsigma+\sigma)^2\tilde{A}^2}{m^2} + \frac{9\alpha^2 n^2\sigma^2}{T} + 21\alpha^2 n^2\frac{1}{T}\sum_{t=1}^{T}\|\nabla f(\boldsymbol{w}_t)\|_2^2,$$

yielding the claim. $\qquad\square$

We next build on Lemma 6.

**Lemma 7.** *If* $\alpha \le \frac{2}{9n\mu}$, *then, for* $\rho = 1 - \frac{\alpha n\mu}{2}$,

$$\sum_{t=1}^{T}\rho^{T-t}\Delta_t^2 \le 12\rho^{T-1}\alpha^2 n^2\sigma^2 + \frac{28\rho\alpha^2(\varsigma+\sigma)^2\tilde{A}^2}{(1-\rho)m^2} + \frac{1}{2L_{2,\infty}^2}\sum_{t=1}^{T}\rho^{T-t}\|\nabla f(\boldsymbol{w}_t)\|_2^2.$$

*Proof.* Recall (15) from Lemma 6:

$$\Delta_1^2 \le 3\alpha^2 n^2\sigma^2 + 3\alpha^2 n^2\|\nabla f(\boldsymbol{w}_1)\|_2^2.$$

We multiply each term $\Delta_t$ with $\rho^{T-t}$ for $t \in [T]$ and get

$$\rho^{T-1}\Delta_1^2 \le \rho^{T-1}3\alpha^2 n^2\sigma^2 + \rho^{T-1}3\alpha^2 n^2\|\nabla f(\boldsymbol{w}_1)\|_2^2. \tag{17}$$

Similarly, recall (16) from Lemma 6,

$$\Delta_{t+1}^2 \le (2/3)\Delta_t^2 + \frac{7\alpha^2(\varsigma+\sigma)^2\tilde{A}^2}{m^2} + 7\alpha^2 n^2\|\nabla f(\boldsymbol{w}_{t+1})\|_2^2,$$

for which we also multiply each term $\Delta_t$ with $\rho^{T-t}$ for $t \in [T]$, and get

$$\rho^{T-t}\Delta_t^2 \leq (2/3)\rho^{T-t}\Delta_{t-1}^2 + \rho^{T-t}\frac{7\alpha^2(\varsigma+\sigma)^2\tilde{A}^2}{m^2} + \rho^{T-t}7\alpha^2n^2\|\nabla f(\boldsymbol{w}_t)\|_2^2$$

$$\leq (3/4)\rho^{T-(t-1)}\Delta_{t-1}^2 + \rho^{T-t}\frac{7\alpha^2(\varsigma+\sigma)^2\tilde{A}^2}{m^2} + \rho^{T-t}7\alpha^2n^2\|\nabla f(\boldsymbol{w}_t)\|_2^2, \quad \forall t \in \{2,...,T\},$$

(18)

where we have used $\alpha \leq \frac{2}{9n\mu}$ so that $\rho = 1 - \frac{\alpha n\mu}{2} \geq (2/3)(4/3)$.

Next, we sum the bounds in (17) and (18) for $\rho^{T-t}\Delta_t$ for all $t \in [T]$, and we get

$$\rho^{T-1}\Delta_1^2 + \sum_{t=2}^{T}\rho^{T-t}\Delta_t^2 \leq \frac{3}{4}\sum_{t=2}^{T}\rho^{T-(t-1)}\Delta_{t-1}^2 + \rho^{T-1}3\alpha^2n^2\sigma^2 + \sum_{t=1}^{T}\rho^{T-t}\frac{7\alpha^2(\varsigma+\sigma)^2\tilde{A}^2}{m^2} +$$

$$7\alpha^2n^2\sum_{t=1}^{T}\rho^{T-t}\|\nabla f(\boldsymbol{w}_t)\|_2^2.$$

We can rewrite the right-hand side as

$$\leq \frac{3}{4}\sum_{t=1}^{T}\rho^{T-t}\Delta_t^2 + \rho^{T-1}3\alpha^2n^2\sigma^2 + \frac{7\rho\alpha^2(\varsigma+\sigma)^2\tilde{A}^2}{(1-\rho)m^2} + 7\alpha^2n^2\sum_{t=1}^{T}\rho^{T-t}\|\nabla f(\boldsymbol{w}_t)\|_2^2$$

$$\leq 12\rho^{T-1}\alpha^2n^2\sigma^2 + \frac{28\rho\alpha^2(\varsigma+\sigma)^2\tilde{A}^2}{(1-\rho)m^2} + 28\alpha^2n^2\sum_{t=1}^{T}\rho^{T-t}\|\nabla f(\boldsymbol{w}_t)\|_2^2.$$

Lastly, we use $\alpha \leq \frac{1}{\sqrt{56}nL_{2,\infty}}$ to get:

$$\sum_{t=1}^{T}\rho^{T-t}\Delta_t^2 \leq 12\rho^{T-1}\alpha^2n^2\sigma^2 + \frac{28\rho\alpha^2(\varsigma+\sigma)^2\tilde{A}^2}{(1-\rho)m^2} + \frac{1}{2L_{2,\infty}^2}\sum_{t=1}^{T}\rho^{T-t}\|\nabla f(\boldsymbol{w}_t)\|^2.$$

$\square$

## C.6 Proof of Theorems 2 and 3

Using the Lemmas above, we next prove our main results, presented in Section 4.

*Proof of Theorem 2.* The given learning rate $\alpha$ satisfies the constraints of Lemma 3 and Lemma 6. Therefore,

$$
\frac{1}{T}\sum_{t=1}^{T}\|\nabla f(\boldsymbol{w}_t)\|_2^2 \leq \frac{2F_1}{\alpha nT} + L_{2,\infty}^2\left(\frac{21(\alpha(\varsigma+\sigma)\tilde{A})^2}{m^2} + \frac{9(\alpha n\sigma)^2}{T} + 21\alpha^2 n^2 \frac{1}{T}\sum_{t=1}^{T}\|\nabla f(\boldsymbol{w}_t)\|_2^2\right)
$$

$$
\leq \frac{4F_1}{\alpha nT} + \frac{42L_{2,\infty}^2(\alpha(\varsigma+\sigma)\tilde{A})^2}{m^2} + \frac{18L_{2,\infty}^2(\alpha n\sigma)^2}{T},
$$

due to $\alpha \leq \frac{1}{\sqrt{42}nL_{2,\infty}}$.

We next derive the convergence rate. Let $\Gamma = \frac{42(L_{2,\infty}(\varsigma+\sigma)\tilde{A})^2}{m^2} + \frac{18L_{2,\infty}^2 n^2\sigma^2}{T}$. Then,

$$
\frac{1}{T}\sum_{t=1}^{T}\|\nabla f(\boldsymbol{w}_t)\|_2^2 \leq \frac{4F_1}{\alpha nT} + \Gamma\alpha^2.
$$

We then set $\alpha \leq \left(\frac{4F_1}{n\Gamma T}\right)^{1/3}$. So we will have $\alpha = \min\left\{\frac{1}{16L_{2,\infty}(2n+\tilde{A}/m)}, \left(\frac{4F_1}{n\Gamma T}\right)^{1/3}\right\}$ or

$$
\frac{1}{\alpha} = \max\left\{16L_{2,\infty}(2n+\tilde{A}/m), \left(\frac{4F_1}{n\Gamma T}\right)^{-1/3}\right\}.
$$

Substitute $\alpha$:

$$
\frac{1}{T}\sum_{t=1}^{T}\|\nabla f(\boldsymbol{w}_t)\|_2^2 \leq \frac{4F_1}{nT}\left\{16L_{2,\infty}(2n+\tilde{A}/m) + \left(\frac{4F_1}{n\Gamma T}\right)^{-1/3}\right\} + \Gamma\left(\frac{4F_1}{n\Gamma T}\right)^{2/3}
$$

$$
\leq \left(\frac{4F_1}{nT}\right)^{2/3}\Gamma^{1/3} + \frac{64F_1 L_{2,\infty}(2+\tilde{A}/(mn))}{T}
$$

$$
\leq \left(\frac{4F_1}{nT}\right)^{2/3}\left(\frac{(\sqrt{42}L_{2,\infty}(\varsigma+\sigma)\tilde{A})^{2/3}}{m^{2/3}} + \frac{(\sqrt{18}L_{2,\infty}n\sigma)^{2/3}}{T^{1/3}}\right)
$$

$$
+ \frac{64F_1 L_{2,\infty}(2+\tilde{A}/(mn))}{T}
$$

$$
\leq \frac{(4\sqrt{42}F_1 L_{2,\infty}(\varsigma+\sigma)\tilde{A})^{2/3}}{(mnT)^{2/3}} + \frac{(72F_1 L_{2,\infty}\sigma)^{2/3}}{T}
$$

$$
+ \frac{64F_1 L_{2,\infty}(2+\tilde{A}/(mn))}{T},
$$

Since $(4\sqrt{42})^{2/3} < 9$, the above is

$$
\leq \frac{9(F_1 L_{2,\infty}(\varsigma+\sigma)\tilde{A})^{2/3}}{(mnT)^{2/3}} + \frac{(72F_1 L_{2,\infty}\sigma)^{2/3} + 64F_1 L_{2,\infty}(2+\tilde{A}/(mn))}{T},
$$

in which the leading term (slowest in terms of $T$) is $\tilde{O}((mnT)^{-2/3})$, proving the claim. $\qquad\square$

*Proof of Theorem 3.* With the P.L. assumption (Assumption 4), we use Lemma 4 and Lemma 7. We show that their constraints are satisfied later) to get

$$F_{T+1} \leq \rho^T F_1 + \frac{\alpha n L_{2,\infty}^2}{2} \sum_{t=1}^{T} \rho^{T-t} \left( \Delta_t^2 - \frac{1}{2L_{2,\infty}^2} \|\nabla f(\boldsymbol{w}_t)\|_2^2 \right)$$

$$\leq \rho^T F_1 + \frac{\alpha n L_{2,\infty}^2}{2} \left( 12\rho^{T-1}\alpha^2 n^2 \sigma^2 + \frac{28\rho\alpha^2(\varsigma+\sigma)^2 \tilde{A}^2}{(1-\rho)m^2} \right)$$

$$\leq \rho^T F_1 + \rho^{T-1} 6\alpha^3 n^3 L_{2,\infty}^2 \sigma^2 + \frac{28\rho\alpha^3 n L_{2\infty}^2(\varsigma+\sigma)^2 \tilde{A}^2}{\alpha n\mu m^2}$$

$$\leq \rho^T F_1 + \rho^T 7\alpha^3 n^3 L_{2,\infty}^2 \sigma^2 + \frac{28\rho\alpha^3 n L_{2\infty}^2(\varsigma+\sigma)^2 \tilde{A}^2}{\alpha n\mu m^2}$$

$$\leq \rho^T (F_1 + \sigma^2/L_{2,\infty}) + \frac{28\alpha^2 L_{2,\infty}^2(\varsigma+\sigma)^2 \tilde{A}^2}{\mu m^2}$$

$$\leq (F_1 + \sigma^2/L_{2,\infty})\exp(-T\alpha n\mu/2) + \frac{28\alpha^2 L_{2,\infty}^2(\varsigma+\sigma)^2 \tilde{A}^2}{\mu m^2},$$

where we have further constrained $\alpha \leq \frac{2}{9n\mu}$ so that $\rho \leq 9/8$ in the forth inequality and $\alpha \leq \frac{1}{7^{1/3}nL_{2,\infty}}$ in the fifth inequality. By setting the derivative w.r.t $\alpha$ of the RHS to 0, the minimizer $\alpha$ under the constraint that $0 < \alpha \leq \min\left\{ \frac{2}{9n\mu}, \frac{1}{16L_{2,\infty}(2n+\tilde{A}/m)} \right\}$ (required by the lemmas) is:

$$\alpha = \frac{2}{Tn\mu} W_0(T^2 m^2 n^2 C_3),$$

as long as

$$T \geq 1 + \frac{2}{n\mu}\max\{(9/2)n\mu, 16L_{2,\infty}(2n+\tilde{A}/m)W_0(T^2 m^2 n^2 C_3)\}$$

$$= 10 + \frac{1}{\mu} 32 L_{2,\infty}(2+\tilde{A}/(mn))W_0(T^2 m^2 n^2 C_3),$$

where $C_3 = \frac{(F_1+\sigma^2/L_{2,\infty})\mu^2}{224L_{2,\infty}^2(\varsigma+\sigma)^2 \tilde{A}^2}$.

What we did here was to set $T$ just large enough so that the minimizer $\alpha$ is the same with or without the constraint.

Denoting $\tilde{W} = W_0(T^2 m^2 n^2 C_3) = \tilde{O}(1)$, we get

$$F_{T+1} \leq \frac{(F_1+\sigma^2/L_{2,\infty})\tilde{W}}{T^2 m^2 n^2 C_3} + \frac{112L_{2,\infty}^2(\varsigma+\sigma)^2 \tilde{A}^2 \tilde{W}^2}{T^2 m^2 n^2 \mu^3}$$

$$\leq \frac{1}{T^2 m^2 n^2} \left( \frac{(F_1+\sigma^2/L_{2,\infty})\tilde{W}}{\tilde{C}_3} + \frac{112L_{2,\infty}^2(\varsigma+\sigma)^2 \tilde{A}^2 \tilde{W}^2}{\mu^3} \right),$$

which shows rate the convergence rate in the P.L. case is $\tilde{O}((mnT)^{-2})$. $\qquad\square$

# D  Experiment Details

Here we provide more extensive details on our empirical results. This includes background information on our experimental setup in the main paper (Appendix D.1), an additional simulation experiment on pre-training and fine-tuning Tiny GPT-2 (Appendix D.2), and an additional simulation experiment that investigates CD-GraB with different learning rates (Appendix D.3). Our source codes can be found here.

## D.1  Additional details on setup for main paper experiments

### D.1.1  Distributed experiments

We provide additional details on the experiments shown in Figure 3.

**Hardware and software.**  We use a single machine with 128 GiB memory, 1 CPU, and 4 Nvidia GeForce 2080ti GPUs for the HMDA mortgage application, M4, and WikiText-2 tasks. We first discard the remainder $N \mod B$, and then randomly partition $n$ to each worker. Our experiments are all implemented with the PyTorch library. We release our code suite at [REDACTED].

**Datasets and models.**

- **Logistic regression on mortgage application (NY 2017 subset)**: The US Home Mortgage Disclose Act (HMDA) makes available US national data regarding mortgage applications, which has recently been packaged up for easy ML research use [7]. We use the binary classification version of the task, which classifies features as either "grant loan" or "deny loan," for the New York (NY) 2017 subset of the dataset, which includes 244107 examples with 18 features. We model this problem using logistic regression, for which we first perform a random 80/20 train/test split on the raw dataset, and then we discard $N \mod B$ ($B$ is the aggregated minibatch size) examples to ensure that each worker receives exactly $n$ examples. We use 1 worker per GPU, and in total we have $m = 4$ workers, and use NCCL [33] as the distributed communication backend; $m = 4, n = 48816, d = 18, B = 16$. We report test accuracy as our evaluation metric.

- **LSTM on WikiText-2**: We follow the settings in Lu et al. [24] and train a 2-layer LSTM with an embedding size of 32 and dropout set to 0. We use backpropagation through time, for which we set the sequence length to 35. We also adopt the word-vector-classifier-weight-sharing strategy inspired by Inan et al. [19]. WikiText-2 [43] has 600 articles in the train set, with more than 2M tokens and 30K vocabulary; the validation and test sets each have 60 articles. We adapt our training script from PyTorch's official Word Language Modeling Github repository. We use 4 workers in total, with each GPU hosting 1 worker, and use NCCL as the distributed communication backend; $m = 4, n = 3728, d = 1081760, B = 16$. We report test perplexity as the evaluation metric, and we follow the HuggingFace's approach of computing perplexity as the exponentiated average negative log-likelihood of a sequence[16].

- **Autoregressive MLP on M4 Weekly Dataset**: We build a 3-layer autoregressive MLP with a hidden dimension of 64. We set input sequence length to be 20 and the output sequence length to be 6. M4 is a time series dataset composed of 100,000 time series for yearly, quarterly, monthly, weekly, daily and hourly data [25], which is drawn from a random sample of ForeDeCk database [42]. We use the weekly data in our experiment. We use 32 workers, where each of the 4 GPUs hosts 8 process workers. We use GLOO as the distributed communication backend. $m = 32, n = 3355, d = 5569, B = 32$. We report test symmetric mean absolute percentage error (SMAPE) as the evaluation metric. We follow the formula of SMAPE in [25] as follows:

$$\text{SMAPE} \triangleq \frac{2}{h} \sum_{t=n+1}^{n+h} \frac{|Y_t - \hat{Y}_t|}{|Y_t| + |\hat{Y}_t|} * 100\%$$

where $Y_t$ is the reference time series value at timestep $t$, $\hat{Y}_t$ is the forecast time series value at timestep $t$, and $h$ is the forecasting horizon and $n$ is the number of datapoints.

**Hyperparameter optimization.**  For all tasks, we tune the learning rate $\alpha$ for D-RR first, and then use the selected learning rate for CD-GraB. Therefore, an performance improvement here implies we would have in-place substitution benefits via switching from D-RR to CD-GraB with identical learning rate and experiment setups. We use SGD with momentum as the optimizer for all tasks. The hyperparameters for each task are as follows:

---

[16]https://huggingface.co/docs/transformers/perplexity

- **Logistic regression on mortgage application (NY 2017 subset)**: $\alpha = 5\text{e-}3 \in \{1\text{e-}2, 5\text{e-}3, 1\text{e-}3\}$, momentum: 0.9, weight decay: 0, $B$: 16.

- **LSTM on WikiText-2**: $\alpha = 5 \in \{5, 10\}$ and decays by 0.1 per 10 epochs, momentum: 0.9, weight decay: 0, $B$: 16.

- **Autoregressive MLP on Weekly M4 Dataset** $\alpha = 1\text{e-}3 \in \{1\text{e-}2, 1\text{e-}3, 1\text{e-}4\}$, momentum: 0.9, weight decay: 0, $B$: 32.

### D.1.2 Memory Overhead of CD-GraB in LSTM on WikiText-2 Task

We profile the CUDA memory usage for the LSTM on WikiText-2 Task with CD-GraB and D-RR to understand the memory overhead of both data permutation algorithms. This memory analysis is both task and implementation dependent, but still serves to illustrate the overarching point that CD-GraB's memory overhead is not so significant. The additional overhead comes from two sources for CD-GraB: communication and example sorting (Figure 8).

In more detail: In our LSTM experiment, each local worker will share its gradients with all other workers at every optimization step. To reduce the communication burden of CD-GraB, we make each local worker function as an order server.[17] The memory consumption of forward, backward, and optimizer states between CD-GraB and D-RR should be (at least approximately) identical. The model size of LSTM is roughly 4 MiB. We use 4 workers, and as each worker (functioning as an order server) needs to *all-gather* gradients, the memory overhead for *all-gather* communication is roughly tensor_size $\times$ # workers = 4 MiB $\times$ 4 = 16 MiB for CD-GraB(we observe 16.51 MiB in practice, Communication in Figure 8), while D-RR only needs to *all-reduce* the gradients (yielding no memory overhead; the communication buffer for *all-reduce* is reusing the same gradient tensor). The PairBalance algorithm (Algorithm 2) internally needs a model-sized accumulator as the running sum $r$, and both computing inner product between $r$ and $g_1 - g_2$ and updating $r$ with $r + sc$ takes virtually no space with a memory-efficient implementation. Therefore, the memory consumption for PairBalance is still roughly 4 MiB (Data Sorter in Figure 8). .

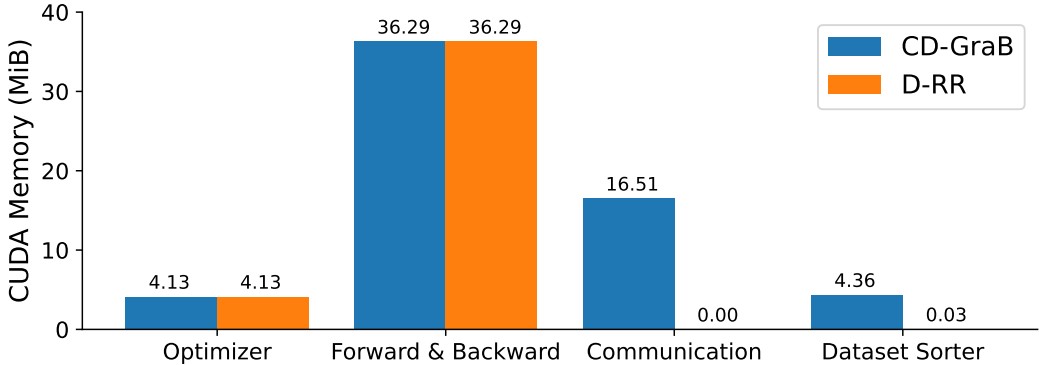

Figure 8: CUDA Memory Overhead of CD-GraB and D-RR in LSTM on WikiText-2 Task.

The main memory overhead of CD-GraB will be dominated by the communication buffer size on the order server side: the order server have to gather the gradient (differences) from all workers, and sequentially apply the PairBalance algorithm. This memory bottleneck would similarly be found in GraB as GraB also needs per-example gradients to perform Balance sequentially.

A future algorithmic improvement to the general gradient balancing framework would be finding a balancing algorithm that does not need per-example gradients to achieve comparable convergence guarantees. However, we still notice that PairBalance is more memory-efficient than Balance as Balance needs to store 3 model-sized tensors: 1 for the balancing accumulator, 1 for running-average gradients for last epoch, and 1 for the running-average for current epoch. In constrast, PairBalance only needs 1 model-sized tensor as the balancing accumulator.

---

[17]An ideal location for a dedicated order server is on a network node that has large input bandwidth and memory buffer to host all gradients while not blocking the normal optimization stages. Since we do not have enough computational resources to host a dedicated order server, we make each worker an order server.

### D.1.3 Simulated ablation study using LeNet on CIFAR-10

In the experiment shown on Figure 4, we select the same learning rate, momentum, and weight decay as the LeNet experiment in Lu et al. [24]. We use 3 different random seeds to control 3 different initialization and the randomness in random reshuffling. The aggregated minibatch size $B$ is 64 for all runs. We implement this ablation study by using 1 GPU with up to $m = 64$ workers (processes). As above, we discard $N \mod B$ examples and partition the remaining examples evenly on each worker.

$\alpha = \text{1e-3} \in \{\text{1e-2, 5e-3, 1e-3, 5e-4, 1e-4}\}$, momentum: 0.9, weight decay: 1e-2, $B$: 64.

We do not implement this via distributed environment due to the fact that we do not have access to 64 GPUs, but expect the simulation results to be a good reflection of the results we would obtain in a multi-GPU setting.

**Parallel herding bound.** We further investigate the empirical parallel herding bounds (8) for the LeNet experiment for the different ordering methods. We plot the results in Figure 9. We observe that as the number of workers increases, the empirical parallel herding bounds of both **ID-GraB (Bal)** and **ID-GraB (PairBal)** also increase, and eventually exhibit little difference with D-RR. CD-GraB, in contrast, exhibits a consistently lower bound.

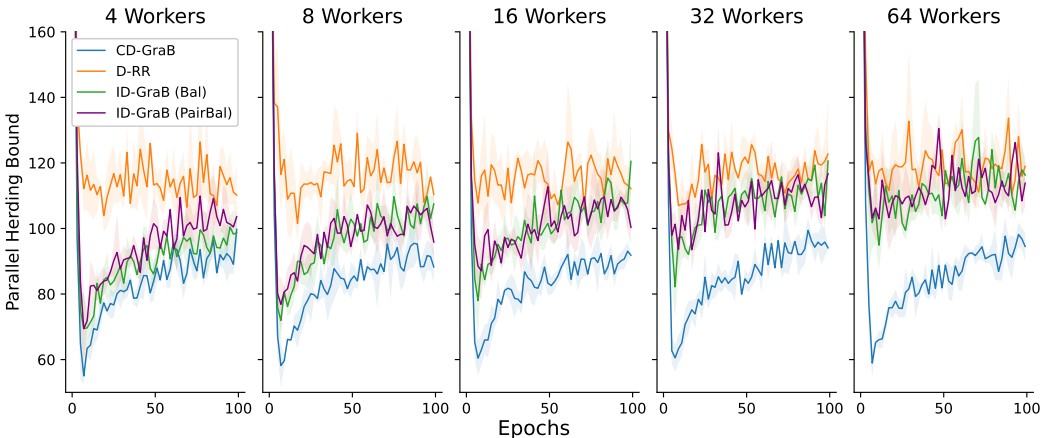

Figure 9: Empirical parallel herding bounds of gradients for each algorithm in LeNet experiment. We plot the mean as the curve and standard deviation across 3 random seeds.

For comparison, we also run a simulation experiment on synthetic data to investigate the behavior of the parallel herding bound. We include these below, in Figure 10.

We randomly initialize 1 million random vectors $z_{i,j}$ from a uniform distribution between 0 and 1 with 16 dimensions as $z_{i,j} \sim \text{Unif}(0,1)^{16}$, and then we zero-center this set of 1 million vectors and normalize them to all have $L_2$ norm as 1. We then evenly partition this set of 1 million random vectors to $\{5, 10, 20, 50, 100\}$ workers and run each example ordering algorithm.

In Figure 10, we run CD-GraB, D-RR, **ID-GraB (Bal)**, **ID-GraB (PairBal)** on these random vectors, and compute the parallel herding bounds (8). From left to right in Figure 10, we observe that as the number of workers $m$ increases, the parallel herding bound of **ID-GraB (Bal)**, **ID-GraB (PairBal)** becomes larger. This shows the importance of coordination when we have a large number of workers.

These results for random vectors cohere with our above results for LeNet on CIFAR-10.

### D.2 An additional simulation experiment: pre-training and fine-tuning Tiny GPT-2

We perform an end-to-end simulation experiment involving pre-training and fine-tuning Tiny GPT-2 on WikiText-103, which we document below.

### D.2.1 Pre-training

We adapt the training script from the HuggingFace's PyTorch casual language modeling code to train the GPT-2 architecture [35]. We set the maximum sequence length to 128 and token and positional

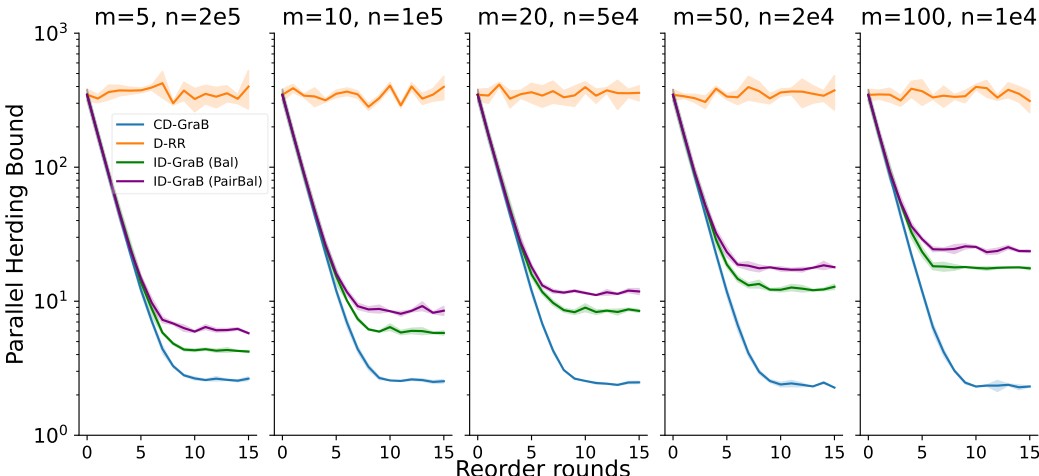

Figure 10: Parallel herding bounds for different example ordering algorithms on $N$=1 million random vectors. We use 3 random seeds, plot the mean and standard deviation across each random seed as the shaded area.

embedding dimension to 128; use 2 hidden layers in the transformer encoder and 2 attention heads; and disable dropout. This model configuration corresponds to the following Python code snippet:

```python
from transformers import GPT2Config, GPT2LMHeadModel, GPT2Tokenizer

tokenizer = GPT2Tokenizer.from_pretrained('gpt2')
config = GPT2Config.from_pretrained('gpt2')
config.n_embd = 128
config.n_ctx = 128
config.n_layer = 2
config.n_head = 2
config.n_positions = 128
config.summary_first_dropout = 0
config.attn_pdrop = 0
config.resid_pdrop = 0
model = GPT2LMHeadModel(config)
```

We train our Tiny GPT-2 model from scratch on WikiText-103 [43]. WikiText-103 is a standard language modeling benchmark that has 28,475 articles in the train set, and 60 for both the validation and test sets, with more than 100M tokens and 267K vocabulary inside the train set. We use the original GPT-2 tokenizer, and use maximum sequence length 128. We note that this is much smaller than the default maximum sequence length for GPT-2, which is 1024, which was too large to use given our computational budget. Nevertheless, 128 is still a reasonable sequence length for the initial phrase of pre-training; BERT uses a sequence length of 128 for the first 90% of pre-training steps to speedup the experiment [9]. We tune the learning rate for D-RR with the grid { 5e-3, 1e-3, 5e-4, 1e-4 } (the final learning rate is 5e-4), and use AdamW optimizer [21]. We use 3 random seeds. Before the training, we simulate 64 workers, and similarly divide the training dataset evenly across them by discarding $N \mod B$ examples. Our hyperparameter optimization space is listed below:

**Pretraining Hyperparameters.** $\alpha$=5e-4 $\in \{$5e-3, 1e-3, 5e-4, 1e-4$\}$, weight decay: 1e-4, $B$: 64.

We document convergence for pre-training in Figure 11, and use test perplexity as our evaluation metric.

### D.2.2 Fine-tuning

We then fine-tune the pre-trained Tiny GPT-2 model on downstream tasks. For each task, we load the pre-trained foundation model weights obtained at the end of 30 epochs of each example ordering algorithm after pretraining, and use the same example ordering algorithm to perform supervised fine-tuning. We focus on the largest 4 GLUE tasks [44]: MNLI, QQP, QNLI, and SST2. We tune the learning rate for D-RR with the AdamW optimizer, and for each run we report the best validation

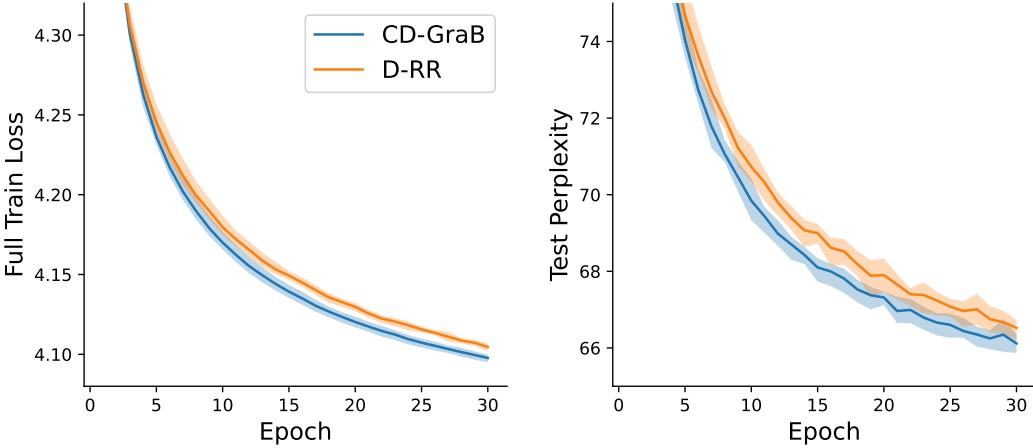

Figure 11: Pre-training Tiny GPT-2 on WikiText-103 from scratch: Convergence for CD-GraB and D-RR with $m = 64$ workers. The aggregated minibatch size per update is 64. We use 3 random seeds, and plot the mean vand standard deviation.

accuracy. We then take an average results of each run and summarize them in Table 1. Our training script is adapted from the HuggingFace's PyTorch GLUE fine-tuning example codes.

**Fine-Tuning Hyperparameters**

- **MNLI** $\alpha = 5\text{e-}4 \in \{5\text{e-}3, 1\text{e-}3, 5\text{e-}4, 1\text{e-}4\}$, Weight decay: 1e-4, $B$: 32, epochs: 10, linear learning rate scheduler

- **QQP** $\alpha = 5\text{e-}4 \in \{5\text{e-}3, 1\text{e-}3, 5\text{e-}4, 1\text{e-}4\}$, Weight decay: 1e-4, $B$: 32, epochs: 10, linear learning rate scheduler

- **QNLI** $\alpha = 5\text{e-}4 \in \{5\text{e-}3, 1\text{e-}3, 5\text{e-}4, 1\text{e-}4\}$, Weight decay: 1e-4, $B$: 32, epochs: 10, linear learning rate scheduler

- **SST2** $\alpha = 5\text{e-}4 \in \{5\text{e-}3, 1\text{e-}3, 5\text{e-}4, 1\text{e-}4\}$, Weight decay: 1e-4, $B$: 32, epochs: 10, linear learning rate scheduler

|          | MNLI (Matched) | MNLI (Mismatched) | QQP | QNLI | SST2 |
|----------|----------------|-------------------|-----|------|------|
| **CD-GraB** | $65.91 \pm 0.46\,\%$ | $64.36 \pm 2.03\,\%$ | $82.25 \pm 0.21\,\%$ | $62.11 \pm 0.70\,\%$ | $82.65 \pm 0.39\,\%$ |
| **D-RR** | $65.42 \pm 0.36\,\%$ | $63.93 \pm 1.63\,\%$ | $81.74 \pm 0.33\,\%$ | $61.87 \pm 0.67\,\%$ | $82.68 \pm 0.57\,\%$ |

Table 1: GLUE fine-tuning datasets: Validation accuracy of CD-GraB in comparison to D-RR, reporting mean and standard deviation of best results for each run. There are 3 runs for each example ordering algorithm.

We include these fine-tuning results in part to support our claim in Section 6 that CD-GraB exhibits its benefits more clearly when there are more training epochs. Our pre-training results suggest that CD-GraB would confer benefits to pre-training large models over multiple epochs; however, CD-GraB will not necessarily be useful for short runs of fine-tuning (as indicated in Table 1, for which the results for both ordering algorithms are effectively identical).

## D.3 Ablation simulation study: The impact of learning rate $\alpha$

In the experiment shown on Figure 12, we select the same momentum and weight decay as the LeNet experiment for 3 random seeds as in Appendix D.1.3. The aggregated minibatch size is still 64 for all runs, and we use 64 workers.

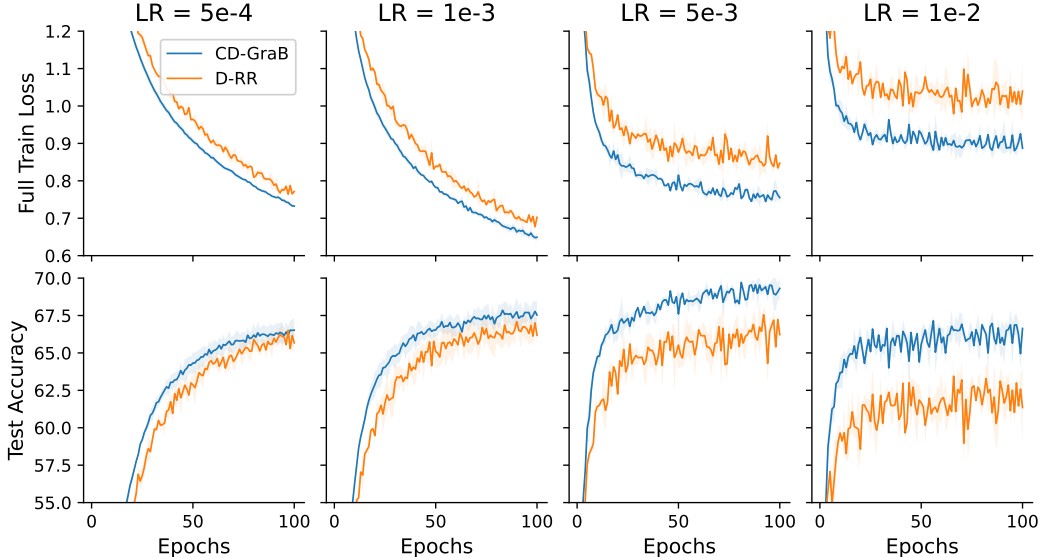

Figure 12: Convergence for CD-GraB, D-RR training LeNet on CIFAR-10, with $m = 64$ workers. The aggregated minibatch size per update is 64. We use 3 random seeds, and plot the mean values across random seeds as the curve, the standard deviation as the shaded area.

We find that when we increase the learning rate from 1e-3 to 1e-2, CD-GraB still maintains relatively better performance than D-RR. The best learning rate for D-RR is 1e-3, in terms of achieving the best test accuracy. We did not tune the learning rate for CD-GraB, and we expect that it is possible to use a higher learning rate and still maintain better empirical performance than D-RR and even faster convergence. We defer such empirical investigations to future work. Altogether, these preliminary empirical results confirm that it is possible to use higher learning rate for CD-GraB, given that online PairBalance does not need to use a stale mean (Section 3.2), which would make larger learning rates perform poorly.