# OpenReview forum: "Coordinating Distributed Example Orders for Provably Accelerated Training"
_NeurIPS.cc/2023/Conference — NeurIPS 2023 poster_

### Official Review · Reviewer_T8gT · 2023-06-24

**Soundness:** 3 good
**Presentation:** 3 good
**Contribution:** 3 good
**Rating:** 6
**Confidence:** 2

**Summary:**

This paper is devoted to the development of the Gradient Balance for the centralized distributed setting.
The proof is highly non-trivial because the extension to distributed ML is not natural. To this end,  the authors use kernel thinning to translate the benefits of provably faster permutation-based example ordering to distributed settings.

**Strengths:**

1. This paper proposes the online Coordinated Distributed Gradient Balancing (CD-GraB) algorithm enabling provably accelerated training in the parallel setting. The new algorithm leverages insights from kernel thinning to elevate the herding framework of centralized GraB (GraB) to the parallel setting.

2. They prove that the convergence rate for CD-GraB exhibits a linear speedup over GraB, which is non-trivial.

3. Empirical results are presented to demonstrate the efficiency of the proposed algorithm.

**Weaknesses:**

1. The presentation may need to be improved. For example, if we turn to  Table 1 （Algorithm 1), the algorithm is not self-explanatory because some notation are not defined.

2. The paper uses a different smoothness assumption. The authors seem to want to hide the degradation from the dimension $d$.

3. The technical difficulty of convergence analysis can be presented as roadmap for the reader.

**Questions:**

What is the main technical difficulty of convergence analysis for distributed GraB.

Minor: line 32: "Gradiant Balance algorithm"  should be "Gradient Balance algorithm"

**Limitations:**

Yes.

---

> ### Author Rebuttal · Authors · 2023-08-10
>
> Thank you for your review. We respond below.
>
> > 1. The presentation may need to be improved. For example, if we turn to Table 1 (Algorithm 1), the algorithm is not self-explanatory because some notation are not defined.
>
> **Algorithm 1**: Thank you for this feedback. Can you please indicate where the presentation is unclear? Which notation did we miss? We will address this in the following discussion period and the subsequent revisions.
>
> > 2. The paper uses a different smoothness assumption. The authors seem to want to hide the degradation from the dimension $d$
>
> Smoothness assumptions follow from the prior work (the original GraB paper) in this space. There is nothing hidden here.
>
> > 3. The technical difficulty of convergence analysis can be presented as roadmap for the reader.
>
> **Proof roadmap**: Thank you for your feedback about this. We provide such a roadmap below, which we will include in revisions of the Appendix.

---

### Official Review · Reviewer_xnQq · 2023-07-06

**Soundness:** 3 good
**Presentation:** 3 good
**Contribution:** 2 fair
**Rating:** 5
**Confidence:** 4

**Summary:**

Propose an online method, CD-GraB for shuffling data samples during distributed model training (parameter-server setting) in a fashion that accelerates convergence in both theory and practice.

Theoretical assumptions are reasonable and quite standard in the literature. Empirically, this work mostly builds on the GraB method, which was described for centralized settings, and extended to the distributed setting in this work.

The difficulty in extending the algorithm to the distributed setting is that
* the considered problem setting only allows workers to produce permutations of their local data samples
* large step sizes results in relatively large disparities across gradients at different workers

**Strengths:**

**Quality**
The work appears to be of a high quality. The proposed method and proofs are non-trivial, and the problem addressed by this work, building on the GraB method, requires careful thought.

**Clarity**
Diagrams are polished, notation is reasonable. The proposed method is non-trivial, and the authors have done a reasonable job at explaining the procedure, with additional illuminating diagrams in the appendix.

**Weaknesses:**

**Significance**
While the overall contribution of the work is impressive, it remains the case that, a) the method is an extension of GraB, and b) not obvious to me that the proposed problem setting is very practical for the considered use cases. It seems to me that much of the difficulty of the analyzed problem is self-imposed: The difficulty in extending the algorithm to the distributed setting is that
* the considered problem setting only allows workers to produce permutations of their local data samples… but often in large-scale DDP model training, data can be read by any worker, and is not stored directly on the computing node. This would enable one to simply extend GraB to such settings.
* while large step sizes may indeed results in large disparities across gradients at different workers… why not AllReduce the gradients, run a gossip step, or simply average the gradients in the parameter server as is typically employed in DDP? In fact, the proposed algorithm already has an aggregation step at the parameter server.

I am also concerned about the overhead of running the PairBalance subroutine in large-scale settings with many workers, especially as the size of model parameters balloons.

On the empirical side, I appreciate the breadth of the exploration, but the improvements training LeNet on CIFAR10 seem quite odd. Why should performance of D-RR change  with the number of workers, when training with the same batch size?

**Questions:**

Please see questions described in-line in weaknesses section.

**Limitations:**

Limitations of the work are not adequately addressed.

---

> ### Author Rebuttal · Authors · 2023-08-10
>
> Thank you for observing the challenging nature of our work and the value of our contributions (both in terms of novel results, and in clarifying them / communicating them effectively). We respond below.
>
> > Significance While the overall contribution of the work is impressive, it remains the case that, a) the method is an extension of GraB, and b) not obvious to me that the proposed problem setting is very practical for the considered use cases. It seems to me that much of the difficulty of the analyzed problem is self-imposed: The difficulty in extending the algorithm to the distributed setting is that the considered problem setting only allows workers to produce permutations of their local data samples… but often in large-scale DDP model training, data can be read by any worker, and is not stored directly on the computing node. This would enable one to simply extend GraB to such settings.
>
> **Problem Setting**: Such an extension of GraB to the distributed setting is an important contribution. GraB is the optimal permutation-based ordering method (see citations within the paper), but does not extend to distributed training and thus is fundamentally limited.
>
> > while large step sizes may indeed results in large disparities across gradients at different workers… why not AllReduce the gradients, run a gossip step, or simply average the gradients in the parameter server as is typically employed in DDP? In fact, the proposed algorithm already has an aggregation step at the parameter server.
>
> **Other DDP approaches** We understand and acknowledge that there exist other solutions to the disparities across gradients in distributed learning, but our solution is orthogonal to theirs. We only consider the problem that how we can find the data permutation for each clients so that their disparities are reduced, and the gossip step approach is not directly related to our solution.
>
> > I am also concerned about the overhead of running the PairBalance subroutine in large-scale settings with many workers, especially as the size of model parameters balloons.
>
> **Memory**: We do not observe overhead issues with the experiments that we could afford to run. We include quantitative results in our response to Reviewer dDcg. As we discuss in our response to reviewer ZrZn, we are actively investigating other ways to approximate gradients as a part of follow-on work to achieve even greater scalability. However, this is out of the scope of the present paper. We can include additional details on limitations and future work in our conclusion (which notes other limitations) in revisions.
>
> > On the empirical side, I appreciate the breadth of the exploration, but the improvements training LeNet on CIFAR10 seem quite odd. Why should performance of D-RR change with the number of workers, when training with the same batch size?
>
> We agree that theoretically D-RR will not be impacted when we use the same batch size in the parameter server side, but in practice, the changes in Figure 4 for D-RR is likely due to the fact that as we change the number of workers, different workers would have a different partition of the dataset.
>
> We also notice that **the difference across the number of workers for D-RR is even less than the standard deviation (the shaded area) when we have the same number of workers** in Figure 4 where we only change the initialization and data permutation of D-RR as we choose multiple seeds. So the fact that the performance of D-RR changes with the number of workers should be of less concern.

---

> > ### Comment · Reviewer_xnQq · 2023-08-21
> > **Re: Rebuttal**
> >
> > I acknowledge having read the author response. I have no concerns with this work from a theoretical standing.  The authors have also addressed my concerns regrading details of the empirical evaluation.
> >
> > My remaining concern is whether the proposed method work has practical applicability in practice for this community, or whether this is closer to an interesting thought experiment. The difficulty of the problem setting still appears self-imposed to me; I acknowledge the authors' response that the distributed setting is important, but in cluster-based workloads, distributed workers read data from a shared file system, and thus are not artificially restricted to local samples. In terms of gradient disparities, which are a fundamental issue in the considered problem setting, there exist simple strategies in the distributed setting for reducing those (e.g., gossip or communication primitives).
> >
> > I will maintain my weak accept recommendation for now and engage in discussion with other reviewers to asses the impact of extending the GraB method to the setting where a) workers can only shuffle local data, and b) gradient disparities across workers can be large.

---

### Official Review · Reviewer_ZrZn · 2023-07-07

**Soundness:** 3 good
**Presentation:** 4 excellent
**Contribution:** 4 excellent
**Rating:** 7
**Confidence:** 4

**Summary:**

The authors propose a distributed version of Grab. The authors both theoretically and experimentally show that there exists a pattern which can improve training convergence of such a method in a distributed setup.

The paper is very well and tightly written with clear limitations. The authors verified their claims both experimentally and theoretically.


**Strengths:**


* The authors have clearly motivated the problem and discussed it in detail how their work compare to other prior works

* The experiments are well executed and clearly demonstrate the benefit of CD-GRAB.

* The authors discussed the proof in detail in the appendix and organized it well.

* All limitations are discussed.


**Weaknesses:**

* As the authors have stated that there is additional memory overhead keeping older gradients. As memory becomes more scarce due to larger models,do authors think there is a way of approximating the previous epoch gradients ?

* The loss for baselines seems a bit high, can authors comment on that, for ex- the PyTorch LSTM example on wikitext-2 gives a loss of around 100, as shown here - https://arxiv.org/abs/1905.13727 why the perplexity score is so far off ?

**Questions:**

Compared to D-RR graphs CD-Graph seems to very smooth. Do the authors have an explanation for this ?

**Limitations:**

The authors have discussed it well.

---

> ### Author Rebuttal · Authors · 2023-08-10
>
> Thank you for your review and questions. Please see our answers below.
>
> > As the authors have stated that there is additional memory overhead keeping older gradients. As memory becomes more scarce due to larger models, do authors think there is a way of approximating the previous epoch gradients?
>
> **Memory overhead, gradient approximation**: We have included quantitative information about the low memory overhead in our response to reviewer dDcg in ML tasks we are afford to run. Overall, the overhead is small for CD-GraB, but we are actively investigating other ways to approximate gradients for example ordering (which will be the subject of future submissions). Our hope is that our next (in-progress) work on this will also no longer require computing per-example gradients. We can include additional details on limitations and future work in our conclusion (which notes other limitations) in revisions.
>
> > The loss for baselines seems a bit high, can authors comment on that, for ex- the PyTorch LSTM example on wikitext-2 gives a loss of around 100, as shown here - https://arxiv.org/abs/1905.13727 why the perplexity score is so far off?
>
> **LSTM loss on wikitext-2**:
> We follow the original GraB paper (except that we tied input embedding matrix weight with the output word classifier (the weight-typing strategy) as indicated in the appendix) to use the embedding dimension as 32. If we use a larger embedding dimension (200 or higher), we could also reach test perplexity below 100.
>
> **Smoothness of CD-GraB results compared to D-RR**:
> The non-smoothness of D-RR is likely due to the variance of stochastic gradients during training, and CD-GraB would reduce the variance of stochastic gradients during training. If we want to make the D-RR training loss curve smoother, we could use a tiny learning rate. We provide an ablation study in Appendix D.3 on the impact from the learning rate and we could observe that D-RR becomes smoother when we decrease the learning rate. CD-GraB, on the other hand, allows the use of a large learning rate to accelerate the training while preserving the final model’s performance. This is a practical advantage of using CD-GraB: stabilizing training with large learning rate would need significantly less optimization steps to reach the same performance.

---

> > ### Comment · Reviewer_ZrZn · 2023-08-14
> > **Thank you**
> >
> > Reading this response and to other reviewers as well, I believe that this is a good paper. Both from theory and practice point of view.
> >
> > I think the writing is measured and clearly lists the potential shortcomings. I feel the authors are forthright and explicit about their shortcomings.
> >
> > I will vote to accept this paper.
> >
> > I have bumped my score to an accept.

---

> > > ### Author Response · Authors · 2023-08-16
> > > **Response to discussion comment by Reviewer ZrZn**
> > >
> > > Thank you so much for participating in the discussion period. We very much appreciate it.

---

### Official Review · Reviewer_AQrA · 2023-07-10

**Soundness:** 3 good
**Presentation:** 3 good
**Contribution:** 3 good
**Rating:** 5
**Confidence:** 4

**Summary:**

The paper introduces Coordinated Distributed GraB, a new method that leverages stale gradients from prior epochs to order examples and achieve a provably faster convergence rate than random reshuffling. CD-GraB demonstrates a linear improvement in convergence rate compared to centralized GraB and showcases superior performance over alternative methods across various benchmark tasks. The contributions of this paper encompass a thorough theoretical analysis of CD-GraB's convergence rate as well as empirical evaluations conducted on three benchmark tasks.

**Strengths:**

1.The paper presents an innovative method called Coordinated Distributed GraB, which builds upon previous work on GraB and kernel thinning. This novel approach capitalizes on the utilization of outdated gradients from prior epochs to prioritize examples, resulting in a demonstrably faster convergence rate compared to random reshuffling.

2.This paper offers a comprehensive theoretical analysis of CD-GraB's convergence rate.  The analysis reveals a linear acceleration in the convergence rate of CD-GraB.

**Weaknesses:**

1. This method is designed specifically for the paradigm involving a parameter server and workers. However, it is important to note that as the number of workers increases, there will be a linear increase in overhead, potentially affecting performance during large-scale training sessions.

2. In comparison to the widely used ring allreduce method for gradient aggregation, the communication of gradients between the parameter server and workers in this method is less efficient. Consequently, the improvement in wall-clock time may not be convincing. Furthermore, the unavailability of the code makes it difficult to reproduce the experiments.

3. The convergence analysis presented in the paper may not be easily applicable to adaptive optimization methods such as AdamW, which is a popular choice for training deep neural networks.

4. It is worth noting that the proposed method is based on stochastic gradients, while it is common practice to train models using minibatch stochastic gradients.

**Questions:**

The theoretical analysis in this paper assumes an ideal scenario where the number of data examples is evenly divisible by the number of workers. However, it is crucial to consider the implications when the numbers cannot be perfectly divided or when workers have different numbers of update iterations. How does CD-GraB address these challenges? Is there any theoretical analysis available to support its handling of such problems?

**Limitations:**

No.

---

> ### Author Rebuttal · Authors · 2023-08-10
>
> Thank you for your thoughtful review and questions,
>
> > 1. This method is designed specifically for the paradigm involving a parameter server and workers. However, it is important to note that as the number of workers increases, there will be a linear increase in overhead, potentially affecting performance during large-scale training sessions.
>
> **Overhead**: We do not observe a meaningful impact for overhead in our distributed experiments or in our simulations. It is of course possible that communication (of gradients) could cause overhead, but we do not observe this to be meaningful in practice for our experiments. As noted in our response to reviewer dDcg, we could not afford to perform larger scale experiments, which we do not believe should be grounds for rejection. Memory overhead is minor for CD-GraB/ PairBalance, which we discuss in the paper.
>
> > 2. In comparison to the widely used ring allreduce method for gradient aggregation, the communication of gradients between the parameter server and workers in this method is less efficient. Consequently, the improvement in wall-clock time may not be convincing. Furthermore, the unavailability of the code makes it difficult to reproduce the experiments.
>
> **Parameter server**: Parameter servers are also a widely-used distributed training paradigm (see cited works in the paper). Our empirical results demonstrate improvements in wall-clock time, as this review notes. We are not sure why this would not be convincing evidence of the success of our method, and we could provide a histogram of forward / backward / communication / dataset permutation computation time per iteration if needed.
>
> > 3. The convergence analysis presented in the paper may not be easily applicable to adaptive optimization methods such as AdamW, which is a popular choice for training deep neural networks.
>
> **Theory and AdamW**: The theory applies to SGD, as the theory for the original GraB algorithm also applies to SGD. Theory for other optimization methods is deferred to future work. We believe that this contribution is itself useful, laying the groundwork for future improvements. This is not a drawback; there is just more research to do in this area in the future. We nevertheless test CD-GraB using AdamW when training a small GPT-2 in the Appendix (which we observe performs well, though does not directly inherit the theory we contribute).
>
> > It is worth noting that the proposed method is based on stochastic gradients, while it is common practice to train models using minibatch stochastic gradients.
>
> **Minibatches**:  Our theory applies to stochastic gradients, just as the original GraB paper does (see their Appendix). As stated in top-level comments A and B, our main contribution is theoretical. Similar to the original GraB paper (see Appendix), we could do additional engineering work to make our approach work for minibatching. This is not something that wouldn’t work for our method, but we don’t believe that this work is necessary for showcasing our main results. Instead, this is work we will include for future systems work that stress-tests all of the systems and implementation efficiency considerations in modern distributed training. Here, we emphasize showing that our method works for per-example gradients in order to map directly onto our problem setup.  Also notice that *when the number of workers is greater than 1, the gradients computed by the central parameter server are also minibatched*. We also consider the setting where the *microbatch per worker is greater than 1* in our logistic regression on mortgage application experiment, where we use 4 workers and a microbatch of 4 examples per worker (so the minibatch per iteration on parameter server aggregation $B$ is 4 * 4 = 16).
>
> > The theoretical analysis in this paper assumes an ideal scenario where the number of data examples is evenly divisible by the number of workers. However, it is crucial to consider the implications when the numbers cannot be perfectly divided or when workers have different numbers of update iterations. How does CD-GraB address these challenges? Is there any theoretical analysis available to support its handling of such problems?
>
> **Ideal scenario**: We discuss this in the paper and in the Appendix. We divide the examples M mod N on the available servers, and discard (at random) the remaining up to N - 1 examples. For large datasets (and even something like N = 64 in the worst case), this should not matter (indeed, we do not observe this to matter empirically). Indeed even in a non-industry scale dataset like CIFAR-10 that has only 50,000 examples, discarding 63 examples in the worst case is still far less than 50,000 and we did not observe a visible performance degradation.

---

### Official Review · Reviewer_dDcg · 2023-07-26

**Soundness:** 3 good
**Presentation:** 3 good
**Contribution:** 2 fair
**Rating:** 5
**Confidence:** 3

**Summary:**

This paper proposes a method to extend the optimal data order finding method GraB to centralized distributed setting. This is achieved by using an alternative balancing method that does not require using stale averages which would cause a problem given higher learning rates in the distributed settings. Furthermore the ordering task is determined by a global coordinator leading to CD-GraB. The effectiveness is demonstrated empirically over random reshuffling as well as running GraB locally on each worker (with and without the alternative balancing method). Finally, an analysis of the convergence rate is given for non-convex smooth functions as well as functions satisfying PL.

**Strengths:**

The proposed alternative balancing method is novel and can be useful both for the single-node and multi-node versions of GraB. On the experiments CD-GraB is faster both in terms of epoch and wall clock time than random reshuffling. Ablation studies clearly show the importance of using the central coordinator. The paper is written clearly and is easy to follow and understand.

**Weaknesses:**

One of the main drawbacks is that the experiment are quite small in scale. The main comparison is using MLPs and logistic regression while the ablation is done using LeNet instead of the larger ResNet models. For the ablation, the number of epochs is also limited and as a result the obtained accuracy is no where near the state of the art. This makes it hard to judge whether the observed improvement will continue to be there as the training goes on further. Since the paper is advocating for using CD-GraB in practice, large-scale evidence is essential. I would also suggest considering the effect of different architectures including Transformers. If for some reason the method is not applicable to some settings this should be clearly discussed.

In addition, in many cases other optimizers such as Adam are used in practice. The possibility of using this method with such optimizers (particularly Adam) as well as its effect on the performance is not investigated which again makes it harder to judge whether it can be directly deployed in practice or not.

Moreover, one thing that I would assume can considerably affect the performance in the distributed setting is the data heterogeneity. However, no discussion on this has been made. It would be nice to have at least one set of experiments in the heterogeneous settings.

As a side note, since the pair balance algorithm can also be applied in the single-node settings, a good comparison to show the effect of this switch is comparing it with the original GraB in the single node setting, both in terms of speed and the best achievable loss. Though this is covered to some extent given that the performance of local-GraB and CD-GraB match with 4 workers (according to Figure 4), I think a direct comparison in the single node setting would still be useful.

**Questions:**

1. Looking at Figure 3, it seems using GraB allows the optimization to reach a better train loss. Is this true or if the normal RR is run for a longer period it finally reaches GraB's loss?

2. How does using CD-GraB affect the learning rate schedule?

**Limitations:**

No specific discussion of limitations is included. I think a quantitative discussion around additional resource requirements (e.g. memory) is needed. Furthermore, as a replacement for the current optimization methods, a discussion about its ability to combine with existing solutions in distributed settings is missing. For example, I am not sure if this method can be combined with existing privacy frameworks which could also highlight a direction for future work.

---

> ### Author Rebuttal · Authors · 2023-08-10
>
> Thank you for your review, and for appreciating the writing quality of our submission. We respond directly to your comments about weaknesses below.
>
> **Scale of experiments**
> * We note in the paper/Appendix that we ran we could afford to. CD-GraB converges faster than D-RR, which the existing results show. We are academics, not an industry lab. We can run LeNet for longer, but saved our training budget for other tasks.
>
> * Based on the intended scope of our contributions (i.e., primarily theory, with empirical implications to be explored in future systems work), the fairest comparison for our experiments is to the original GraB paper (also a theory-focused paper). We used this prior work as the basis for designing our experiments.
>
> * We additionally include results for a small GPT-2 on WikiText-103 in the Appendix. We note in the conclusion why we did not focus on transformers. Models like LLMs today are finetuned typically for a really small number of epochs. The benefits of CD-GraB (i.e., getting the theory-backed benefits of achieving the herding bound) only come in the **multi-epoch** setting, and we cannot afford the pretraining costs of a LLM for a lot of epochs.
>
> * Even so, we demonstrate the benefits of CD-GraB on other ML tasks, and believe that this is an important contribution. It is worth investing the effort in future ML systems research to build and stress-test a full implementation of an order server at scale. Without the present work, we would not have been able to know that such an investment was worth it. Such a system necessarily is the realm of future work.
>
> **Adam**: The theoretical guarantees for GraB (upon which we base CD-GraB) are for SGD. So it makes sense to run our experiments using SGD. We can provide more details about this context in revisions. This is another reason also we de-emphasize our GPT-2 experiment, which uses AdamW for pretraining and finetuning.
>
> **Data heterogeneity:** Data heterogeneity is common in federated learning. We state in footnote 5 that our formulation is not designed for federated learning; it is not in scope. If we were to allow for such data organization, we would need to assume non-global communication per iteration or additional constraints on how global communication occurs.
>
> **Single node pair-balance:** The point of our method is to coordinate and distribute computation where we have >1 worker. We can add this to our Appendix, but do not believe this is important for show-casing our main results.
>
> **D-RR v. GraB:** GraB often performs better than RR in practice (see also the original GraB paper), and in cases like LSTM on WikiText-2 is is clear that running RR longer still cannot reach GraB's training loss, and this is demonstrated both in our paper and the original GraB paper.
>
> **Learning rate schedule**: We used standard settings for the tasks that we investigated, and for the LSTM on WikiText-2 experiment we still used learning rate scheduler. Investigating how to tune the learning rate schedule in response to our improved convergence is the realm for future theory work.
>
> **Memory:** We believe that the extra memory requirements follow directly from our algorithm, so we cut the histogram of LSTM on WikiText-2 memory for space, which is associated with the Figure 3.b. We need 1 copy of the model parameters as the accumulator in the order server, and both obtaining a pair gradient element-wise difference and calculating inner product computation during the dataset sorting step is O(1). The accumulator update is in-place. Notice that in experiments we always place the order server in the GPU, but we could further alleviate the memory concern by moving it to the CPU as getting next-epoch permutation is not blocking any of the training stages in GPU.
>
> | Memory Overhead (MiB) for LSTM on WikiText-2 | Optimizer | Forward & Backward | Communication | Dataset Sorter
> | :---        | :----: |   :---: | :---: | :---: |
> | CD-GraB  | 4.13   | 36.29   | 16.51 |  4.36 |
> | D-RR        | 4.13   | 36.29   |  0.00 |  0.03 |
>
> **Overall comment about review and project scope:** Given the length of conference submission, we don’t have much more room for experiments than the quantity that we already present. Our main contribution, which fills the majority of our paper and Appendix, is theoretical. To include 3 additional sets of experiments (as requested) would require us to downplay our main contribution. We aren’t claiming that our work is a general-purpose solution for all ML (e.g., privacy-preserving frameworks). We situate our discussion in relation to parameter-server architectures and the distributed training setup. How our work could be combined with privacy techniques is the subject of another paper.
>
> **Request to please consider revising score**:
> The review acknowledges that we’ve made a sound and novel theory-backed contribution that we support with experiments. We understand that our experiments are not the largest scale that they could be, but we ran what we could afford to and believe that the experiments we ran are appropriate for the contributions we are making. They are in line with and go beyond what GraB did.
> The emphasis on empirics in this review does not address the value of our theoretical contributions. Given the length of a conference paper, to make all of the additions that this review requests (which, as discussed above, we don’t believe are entirely appropriate for the present work) would require us to significantly cut down on our discussion of our main contribution. This feedback is helpful for us to consider when scoping out future work.
> Nevertheless, given the novelty and promising nature of our results, our theory contributions, and how our work suggests that it is worth investing future effort in building a large scale system (which we could not have known was worth doing before the present work), we believe such a low score (3/rejection) is quite harsh.
>
> Thank you for your consideration.

---

> > ### Comment · Reviewer_dDcg · 2023-08-15
> >
> > Dear Authors,
> >
> > Thank you for your replies.
> >
> > > We note in the paper/Appendix that we ran we could afford to. CD-GraB converges faster than D-RR, which the existing results show. We are academics, not an industry lab. We can run LeNet for longer, but saved our training budget for other tasks.
> >
> > I completely understand the challenge of limited resources. However, there exists many prior works from academia applying larger models such as ResNet on CIFAR10. I point out that there has been cases where proposed optimization methods fail to reach the high accuracy obtained by random reshuffling GD even though they might be faster in the beginning of training. Therefore, plots showing random reshuffling is outperformed by CD-GraB can become misleading especially for readers not familiar with the expected state of the art results.  This is why I find the evidence provided here very insufficient to support practical use of CD-GraB.
> >
> > > Based on the intended scope of our contributions (i.e., primarily theory, with empirical implications to be explored in future systems work), the fairest comparison for our experiments is to the original GraB paper (also a theory-focused paper). We used this prior work as the basis for designing our experiments.
> >
> > In the whole paper, the argument has been always about working both in theory **and** in practice. If this is not the case, this should be clarified and the reason why the optimizer is not practice-ready explained.
> >
> > > We additionally include results for a small GPT-2 on WikiText-103 in the Appendix. We note in the conclusion why we did not focus on transformers. Models like LLMs today are finetuned typically for a really small number of epochs. The benefits of CD-GraB (i.e., getting the theory-backed benefits of achieving the herding bound) only come in the multi-epoch setting, and we cannot afford the pretraining costs of a LLM for a lot of epochs.
> >
> > Thank you. Please add a reference to these experiments in the main text as well.
> >
> > > Even so, we demonstrate the benefits of CD-GraB on other ML tasks, and believe that this is an important contribution. It is worth investing the effort in future ML systems research to build and stress-test a full implementation of an order server at scale. Without the present work, we would not have been able to know that such an investment was worth it. Such a system necessarily is the realm of future work.
> >
> > If I understand correctly, this means that there exist a bottleneck or an obstacle in scaling up CD-GraB which needs a full implementation. This is what I was asking for in the limitation section of my original review. What are these limitations? It does not seem to be the memory based on your explanation. Assuming off-loading to CPU is not needed, what other significant implementation is needed or what part of the algorithm needs to be improved to make it suitable for large scale experiments?
> >
> > > Single node pair-balance
> >
> > Please do add the experiments if you have them. If I am understanding correctly, replacing the ordering method can be beneficial even in the single node setting since it removes the need to store all past gradients. It would be also important to assess whether using this method instead of the original method is less performant in finding the right order or not (which might be hidden in the multi node settings due to incompatibility of the original method with that setting). In general, unless needed, changing only one hyper-parameter (algorithm) instead of both (algorithm and number of workers) allows for better comparison.
> >
> > > Learning rate schedule: We used standard settings for the tasks that we investigated, and for the LSTM on WikiText-2 experiment we still used learning rate scheduler. Investigating how to tune the learning rate schedule in response to our improved convergence is the realm for future theory work.
> >
> > I am not sure what is the standard schedule for LeNet on CIFAR10. A decay in learning rate usually shows itself in a spike in accuracy but I could not find this in the plots. Can you please describe the standard schedule you mentioned?
> >
> > **Regarding the memory**: Thank you for the explanation. I did not understand what is exactly reported in the table. What is meant by overhead? What is the increase in full memory usage (percentage for example)?
> >
> > **Overall comment and suggestion**: My main objection currently is that the evidence presented is not convincing enough for using this method directly in practice which I initially considered to be part of the claim. Still the idea behind the paper is novel and as the authors said the method can open doors for further improvements. My suggestion is that the authors would add a Limitation section clearly discussing all the limitations including the gap between the accuracies on CIFAR10 and state of the art, possible bottlenecks if any (discussed above), etc. to avoid any confusion. Assuming this is possible (please confirm), I have updated my score.
> >
> > Thank you.

---

> > > ### Author Response · Authors · 2023-08-16
> > > **Response to discussion comment by Reviewer dDcg**
> > >
> > > Thank you so much for participating in the discussion period. We respond below:
> > > > I point out that there has been cases where proposed optimization methods fail to reach the high accuracy obtained by random reshuffling GD even though they might be faster in the beginning of training. Therefore, plots showing random reshuffling is outperformed by CD-GraB can become misleading especially for readers not familiar with the expected state of the art results. This is why I find the evidence provided here very insufficient to support practical use of CD-GraB.
> > >
> > > Thank you for clarifying your initial comment. We observe that we often perform *better* than D-RR, not just at the beginning of training, but overall. If accepted, we will update with revised results/figures that make this clearer. Unfortunately, we cannot do so in the discussion period.
> > >
> > > > In the whole paper, the argument has been always about working both in theory and in practice. If this is not the case, this should be clarified and the reason why the optimizer is not practice-ready explained.
> > >
> > > We will update the paper to reflect this. The paper is a theory-based study, which we validate empirically. We scoped it this way to determine the viability of building a fully implemented distributed system. Such a system would be needed to showcase how others more generally could use CD-GraB in practice (i.e., doing distributed training across multiple nodes, if the application warranted this scale, not just within a node with multiple workers).
> > >
> > > > Please add a reference to these experiments in the main text as well.
> > >
> > > If accepted, this is what we would use the additional page for.
> > >
> > > > this means that there exist a bottleneck or an obstacle in scaling up CD-GraB which needs a full implementation. This is what I was asking for in the limitation section of my original review. What are these limitations? It does not seem to be the memory based on your explanation. Assuming off-loading to CPU is not needed, what other significant implementation is needed or what part of the algorithm needs to be improved to make it suitable for large scale experiments?
> > >
> > > It isn't a bottleneck so much as it requires multiple months of software engineering for a full distributed system implementation. We want to do this work, since we now know from the success of the present work that it is worth the investment to build such a system. We are happy to provide more context on the work needed for such a system in the Appendix.
> > >
> > > > Please do add the experiments if you have them. If I am understanding correctly, replacing the ordering method can be beneficial even in the single node setting since it removes the need to store all past gradients. It would be also important to assess whether using this method instead of the original method is less performant in finding the right order or not (which might be hidden in the multi node settings due to incompatibility of the original method with that setting). In general, unless needed, changing only one hyper-parameter (algorithm) instead of both (algorithm and number of workers) allows for better comparison.
> > >
> > > Yes, it is beneficial. We will include these results (we have them from our initial testing of PairBalance, just deleted them due to space).
> > >
> > > > I am not sure what is the standard schedule for LeNet on CIFAR10. A decay in learning rate usually shows itself in a spike in accuracy but I could not find this in the plots. Can you please describe the standard schedule you mentioned?
> > >
> > > We used the settings from the original GraB paper (which were in turn chosen from prior work). We will make these choices clearer int he Appendix.
> > >
> > > > Thank you for the explanation. I did not understand what is exactly reported in the table. What is meant by overhead? What is the increase in full memory usage (percentage for example)?
> > >
> > > We apologize for this. The rebuttal window closed while we were still making unification edits to our responses. Please refer to the top-level response to all reviewers for a figure (attached as a PDF) that more clearly captures this table. We noticed on the late side that we could include such a PDF with a figure.
> > >
> > > > My main objection currently is that the evidence presented is not convincing enough for using this method directly in practice which I initially considered to be part of the claim. Still the idea behind the paper is novel and as the authors said the method can open doors for further improvements. My suggestion is that the authors would add a Limitation section clearly discussing all the limitations including the gap between the accuracies on CIFAR10 and state of the art, possible bottlenecks if any (discussed above), etc. to avoid any confusion. Assuming this is possible (please confirm), I have updated my score.
> > >
> > > We hope we have appropriately confirmed in our answers above. We will also more clearly call out the limitations that we mention in the conclusion using a bolded paragraph heading.
> > >
> > > Thank you again

---

### Author Rebuttal · Authors · 2023-08-10



# Top-level response to all reviewers

Thank you for your comments. We appreciate the detailed feedback, and think that there are ideas in the reviews that would be interesting to pursue in future work (but unfortunately don’t fall in scope for the present project). We clarify below.

## A. Prior work, our work, & future work

We clarify how our work fits with prior work, what we intend to (and do) solve, and what this enables for future work.

### i. Prior
Provably better permutation-based example orderings **for SGD** is a relatively new area. We consider there to be 3 papers (starting in 2022) that directly precede ours:
1. “A General Analysis of Example-Selection for Stochastic Gradient Descent” provides a framework for reasoning about example order and **SGD**

2. “GraB” finds provably better orders **for SGD** in the **single node** case

3. “Tighter Lower Bounds for Shuffling SGD Random Permutations and Beyond” proves that GraB is optimal (for SGD, single node)

**None of this work applies to the modern setting of distributed training.** It assumes local access to all examples, which isn't the case in popular distributed settings (e.g., parameter server with workers with access to subsets of examples).

### ii. Our work
Our main contribution is to prove essential theory that lifts this promising prior work to the distributed setting. **Our main contribution is theoretical.** We make GraB work in the distributed setting (i.e., better permutation-based example ordering for **SGD**) and guarantee a linear speed-up in the number of distributed workers. We solved the problem we set out to solve (and verified our solution with experiments **for SGD**). Based on our reading of the reviews, the novelty and role of our theory is not in dispute (and this is our main contribution).

### iii. Future work
Our theory has direct empirical implications for future work. We'd like to build a distributed ML system, which has an order server, parameter server(s), and workers that efficiently orchestrate distributed example selection. **But to know that it was worth building such a system, we first had to first make the contributions in this paper.** We had to:

1. Prove that it was possible to make GraB work in the distributed setting (which we do); and,
2. Verify empirically that we're able to see improvements in convergence time (which we also do).

These two aims more than fill a conference-length paper (we have a large Appendix). Contributions beyond this are out of scope, e.g.:

* **Systems considerations**: Future work should address implementation optimizations, the engineering particulars of minibatching and CD-GraB (also see the original GraB paper’s Appendix for a sketch of how such an implementation would work), full system-level profiling of our method in comparison to other balancing strategies, detailed HPO, etc. **Such analysis would be significantly more useful for such a fully-implemented ML systems paper, which is not what our present paper is about.**

* **Other optimizers**:  It'd be interesting to study example ordering for Adam and other optimizers (both single node and distributed). These are contributions for other papers, not the present one. E.g., Single-node guarantees for Adam would be a different way to extend prior work, but that isn't the direction we took; we solved something else. This is also why our experiments in the main paper involve **SGD**, not other optimizers; see Appendix for additional experiments.

## B. We ran the experiments we could afford to

Since our main contribution is theoretical (see A.ii), **our empirical contributions are secondary**. We'll adjust the paper presentation to make this clearer (this change is cosmetic, not substantive). As academics, we had a limited training budget for this project, which we exhausted for a project with these theoretical aims. Further, larger scale stress-testing of distributed example selection requires future research investment to build a working distributed ML system, which we intend to budget for and do. **But first**, we needed to make sure our approach was theoretically sound --- to justify making such an investment in systems research in this direction (see A.iii). **Our experiments achieve this aim: They show that our theory works in practice; they validate that it'd be worth investing significant resources in building such a system and testing it at a much larger scale.**

## C. Our empirical work should use GraB as the basis for comparison
For the reasons above (theoretical aims, relationship to prior work), **our experiments should be evaluated in comparison to what the original GraB paper did**. We made our experiments consistent with those the original authors ran, **and we did more** (including larger scale, see our Appendix). We believe that this is the fair comparison for what was sufficient to stress-test our theoretical contributions.

## D. Memory footprint results

Memory requirements follow directly from our algorithm: We need 1 copy of the model parameters as the accumulator in the order server; both obtaining a pair gradient element-wise difference (could be done in-place) and calculating inner product computation during the sort step is O(1). The accumulator update is in-place so there is no memory overhead. We cut figures for space, but will include in revisions, e.g., for LSTM on Wikitext2 (Figure 3b), see the attached PDF. The memory bottleneck is more relevant when it comes to manifesting and communicating the per-example gradients on the order-server side. However, notice this is still a minor concern because **clients only need to know the next epoch data permutation at the end of the epoch**. We could place the order server in CPU and ***offload the gradients in CPU RAM**, which usually has much larger memory than GPU (we didn't do this for simplicity of our implementation, but could, as dataset sorting is not blocking any of the forward / backward / optimization stages).

---

### Decision · Program_Chairs · 2023-09-21

**Decision:**

Accept (poster)

**Comment:**

This paper proposes the Coordinated Distributed Gradient Balancing method (CD-GraB) to accelerate training in the distributed setting. The theoretical analysis is provided for general smooth nonconvex functions and functions with the PL condition. Experimental results on logistic regression, LSTM, and autoregressive MLP show that the proposed method improves over the distributed random reshuffling (D-RR). Reviewers unanimously recommended accepting this paper and all concerns were well-addressed in the rebuttal. Therefore I recommend accepting this paper.